# `TopoSRL`: Topology Preserving Self-Supervised Simplicial Representation Learning

**Hiren Madhu and Sundeep Prabhakar Chepuri**
Indian Insitute of Science
`hirenmadhu, spchepuri@iisc.ac.in`

## Abstract

In this paper, we introduce `TopoSRL`, a novel self-supervised learning (SSL) method for simplicial complexes to effectively capture higher-order interactions and preserve topology in the learned representations. `TopoSRL` addresses the limitations of existing graph-based SSL methods that typically concentrate on pairwise relationships, neglecting long-range dependencies crucial to capturing topological information. We propose a new simplicial augmentation technique that generates two views of the simplicial complex that enriches the representations while being efficient. Next, we propose a new simplicial contrastive loss function that contrasts the generated simplices to preserve local and global information present in the simplicial complexes. Extensive experimental results demonstrate the superior performance of `TopoSRL` compared to state-of-the-art graph SSL techniques and supervised simplicial neural models across various datasets corroborating the efficacy of `TopoSRL` in processing simplicial complex data in a self-supervised setting.

## 1 Introduction

Simplicial complexes are mathematical structures that explicitly capture higher-order relationships between entities (as nodes) using simplices of different orders (as edges, triangles, and so on). There has been a growing interest in developing simplicial representation learning models, such as simplicial neural networks (SNN) [1–6], as simplicial complexes are generally more expressive than graphs, which only capture pairwise relations. SNNs incorporate topological information available in simplicial complexes while learning representations of simplices of different orders, which are useful for various downstream tasks such as node, graph, higher-order link, and trajectory prediction tasks [1, 4–6]. However, a significant challenge in learning representations for simplicial complexes using existing SNN models is the need for task-specific labels required for training. Acquiring meaningful labels for real-world, high-dimensional, and complex data is difficult due to intricate structures, multiple valid labeling schemes, or privacy and ethical concerns.

Self-supervised learning (SSL) schemes learn expressive and powerful representations without requiring labeled data. Specifically, the main goal in SSL is to model an encoder, which is learned using an objective function and unlabelled training data. This paper proposes an SSL method for simplicial complex data that preserves topological and geometric information while learning representations. Although no existing studies focus on SSL for simplicial complex data, a closely related field of SSL for graph data has been extensively studied. SSL for simplicial complex data is an important generalization as every simplicial complex inherently includes an underlying graph, making SSL on graphs a specialized subset of SSL on simplicial complexes.

The general idea behind SSL on graphs is to augment a graph to create two views of the available graph and then maximize the mutual information (MI) between the augmented graphs. So, the research focus thus far has been on designing augmentation techniques and objective functions that maximize MI. However, existing methods of SSL on graphs require complex augmentation [7], negative

37th Conference on Neural Information Processing Systems (NeurIPS 2023).

sampling algorithms [8], or require components to empirically avoid degenerative solutions [9–12]. This leads to a more complex neural model, hindering their direct extension for SSL on simplicial complexes. For instance, deep graph infomax (DGI) [9] maximizes MI between a node and its subgraph by learning two MLPs: one for the subgraph readout and another as a discriminator function classifying if a node exists in the given subgraph or not. Extending DGI to simplicial complexes would necessitate training $k + 1$ discriminators for a simplicial complex of order $k$ (see Section 3 for the definition of the order of a simplicial complex), drastically increasing the training complexity. Similarly, graph contrastive representation learning (GRACE) [8] and graph contrastive representation learning with adaptive augmentation (GCA) [7] need the selection of effective negative samples and require additional storage by selecting all the other nodes in the graph as negative samples. Directly extending these methods to simplicial complexes would lead to prohibitive computational complexity going up to $\mathcal{O}(2^N)$ for a simplicial complex with $N$ nodes. To mitigate the need for negative samples or additional networks, recent approaches like bootstrapped graph representation learning (BGRL) [11], SelfGNN [12], and canonical correlation analysis inspired self-supervised learning on graphs (CCA-SSG) [13] attempt to learn representations by contrasting corresponding node pairs in the two augmented graphs. Self-supervised masked graph autoencoders (GraphMAE) [14] is an alternative approach that uses a reconstruction loss rather than the commonly used contrastive loss in the graph-based SSL methods. However, these methods focus only on contrasting or reconstructing local information and do not account for global long-distance information available in the network.

Motivated by the aforementioned limitations of SSL methods on graphs, we introduce `TopoSRL`, a self-supervised learning pipeline for simplicial complex data that preserves topology information in the representation space. Preserving topology information is crucial because it allows the learned representations to capture higher-order interactions and relationships unique to simplicial complexes. `TopoSRL` comprises an intuitive technique to generate stochastically augmented views of a simplicial complex. We introduce a new contrastive loss function to preserve topology information of simplicial complexes in the geometric space to learn more expressive representations. In sum, our major contributions in `TopoSRL` are as follows:

- **Simplicial augmentation:** We introduce a simple and effective augmentation technique for simplicial complexes that captures the inherent relationships between simplices. In particular, the augmentation method stochastically removes closed simplices and adds open simplicies guided by the topological structure of a simplicial complex, where a closed simplex is a simplex that is part of the simplicial complex, while an open simplex is a simplex that itself is not a part of the simplicial complex, but all of its subsets are. This procedure is computationally efficient and leads to superior performance than randomly adding simplicies without accounting for the topological structure of a simplicial complex.

- **Simplicial contrastive loss:** We propose a loss function for SSL on simplicial complexes that contrasts pairs of corresponding simplices as well as considers the relational distance between pairs of simplices and their augmented counterparts, where the relational distance refers to a measure of dissimilarity between pairs of simplices. We also provide theoretical evidence that the proposed loss function implicitly maximizes MI between a simplex and its neighborhood within the same augmented simplex and the other augmented simplex, helping the model capture the inherent structures and patterns in the data, thereby improving the model's performance and adaptability to various tasks without the requirement of training additional components such as MI estimators.

We conduct experiments demonstrating our proposed method on downstream tasks such as node classification, simplicial closure, graph classification, and trajectory prediction. We also highlight the effectiveness of TopoSRL in learning expressive representations for simplicial complexes with the proposed augmentation technique as opposed to the random augmentation technique. Experiments show that without any complex architectures or expensive augmentation techniques, our method outperforms existing state-of-the-art graph representation learning methods while being competitive with supervised simplicial representation learning methods.

## 2  Related works

This section briefly discusses a few popular SNN models, one of which can be used in `TopoSRL`. The first step of `TopoSRL`'s learning phase is simplicial augmentation. Hence, we also discuss existing

augmentation techniques used in graph SSL and how they can not be extended (due to computationally or empirical limitations) for simplicial complexes. Then, we move on to discussing SSL on graphs, a specialized version of SSL on simplicial complexes.

**Simplicial representation learning.** SNN models learn representations of simplices of different orders (e.g., nodes, edges, triangles, and so on) in a simplicial complex [2, 3], and are based on an extension of graph convolutions to convolutions over simplicial complexes. Simplicial attention networks (SAN) [1] extends graph attention networks (GAT) [15] for simplicial complexes. Message passing simplicial networks (MPSN) [5] proposes a framework to design more expressive SNN models related to the so-called simplicial Weisfeiler-Lehman isomorphism test. `TopoSRL` is free from the choice of a specific SNN encoder, and any one of the SSN models can be used as an encoder.

**Graph augmentation.** Most of the graph SSL methods described in Section 1, namely, GRACE, GCA, and BGRL, use an augmentation technique that includes two steps: (i) add and remove edges at random to generate two graphs from an input graph, and (ii) randomly mask dimensions of initial node features. GCA [7] proposes an adaptive data augmentation technique that models the edge-removal probability differently for each edge in the graph based on the importance of the edge. However, this method requires more expensive augmentations to attain peak performance [11]. Applying an identical augmentation technique to simplicial complexes necessitates modeling removal probabilities for each simplex. This increases the augmentation process's complexity and mandates the development of innovative metrics for measuring simplex importance. Other than this, most methods use uniform probability to add and remove edges. Although adding and removing simplicies uniformly at random is a naive and simple extension, we empirically show that TopoSRL performs considerably better than random augmentation. In contrast, `TopoSRL` uses a simpler augmentation technique that only requires the addition of open simplicies, which are inherently rich in information as discussed later in Section 4.1, and is very easy to implement.

**SSL on graphs.** Contrastive learning methods for images have recently been adapted for graphs. This includes DGI [9], which contrasts node-local patches with global graph representations was inspired by Deep InfoMax [16]. GMI [17] maximizes a concept of graphical mutual information inspired by MINE [18], enabling a more granular contrastive loss than DGI. GRACE [8] and its derivatives, such as GCA [7], which rely on more complex data adaptive enhancements, have adapted the SimCLR [19] algorithm for graphs. GraphCL [20] also adapts SimCLR to learn graph-level representations with a contrastive objective. Multi-view graph representation learning (MVGRL) [10] extends contrastive multi-view coding to graphs. All these methods suffer from considerable computational complexities because they rely heavily on negative samples. BGRL [11] and SelfGNN [12] extend Bootstrap you own latent (BYOL) [21], which uses different online and target encoders, wherein the target encoder is updated as an exponential moving average of the online encoder while the online encoder is updated by optimizing a loss function. The difference between BGRL and SelfGNN is that BGRL uses different online and target encoders, but SelfGNN utilizes the same encoder as online and target. Lastly, CCA-SSG[13] uses CCA-based loss. BGRL, SelfGNN, and CCA-SSG are methods free from negative samples and incur lower computational complexity than previous methods. Inspired by these graph SSL methods, `TopoSRL` aims at a negative sampling-free approach focusing on preserving topology and implicitly maximizing MI.

## 3 Background

A simplicial complex $\mathcal{X}$ is a collection of a finite number of simplicies. A simplex of order $k$ (or, a $k$-simplex) is a $(k+1)$-cardinality subset of the vertex set $\mathcal{V}$ so that if a simplex $\sigma_k \in \mathcal{X}$ then all the non-empty subsets of $\sigma_k$ also belong to the simplicial complex $\mathcal{X}$. The order, $K$, of a simplicial complex $\mathcal{X}$ is the order of the maximally ordered simplex in the simplicial complex.

Suppose $\sigma_k$, $\tau_k$ and $\rho_k$ are some $k$-simplices in a simplicial complex. A $k$-simplex $\sigma_k$ has the following neighbors in the simplicial complex, namely, lower-adjacent neighbors $\mathcal{L}(\sigma_k) = \{\tau_k | \rho_{k-1} \subset \sigma_k \wedge \rho_{k-1} \subset \tau_k\}$, upper-adjacent neighbors $\mathcal{U}(\sigma_k) = \{\tau_k | \sigma_k \subset \rho_{k+1} \wedge \tau_k \subset \rho_{k+1}\}$, boundary $\mathcal{B}(\sigma_k) = \{\tau_{k-1} | \tau_{k-1} \subset \sigma_k\}$, and co-boundary $\mathcal{C}(\sigma_k) = \{\tau_{k+1} | \sigma_k \subset \tau_{k+1}\}$.

An *open $k$-simplex* is defined as a $k$-simplex $\sigma_k$ such that all its boundaries $B(\sigma_k) \subset \mathcal{X}$, but $\sigma_k \notin \mathcal{X}$. In other words, if all the boundaries of a simplex are present in the simplicial complex, but the simplex itself is not part of the simplicial complex, it is considered an open simplex.

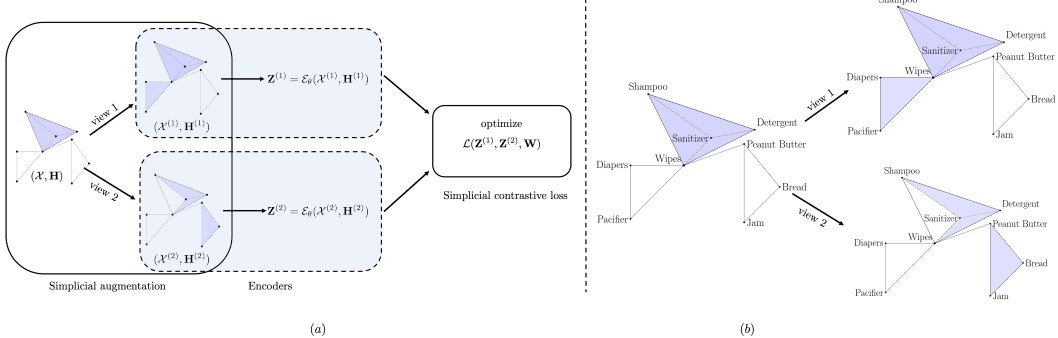

Figure 1: (a) The `TopoSRL` pipeline. Here, $\mathbf{W}$ is a weight matrix that emphasizes the relations between simplices in the two augmented simplicial complexes. (b) Simplicial augmentation example.

A generic SNN model has the following aggregation steps

$$m_B^{t+1}(\sigma_k) = \texttt{AGGREGATE}(\phi_B(h_{\sigma_k}^t, h_{\tau_{k-1}}^t, \forall \tau_{k-1} \in \mathcal{B}(\sigma_k))$$

$$m_C^{t+1}(\sigma_k) = \texttt{AGGREGATE}(\phi_C(h_{\sigma_k}^t, h_{\tau_{k+1}}^t, \forall \tau_{k+1} \in \mathcal{C}(\sigma_k)))$$

$$m_L^{t+1}(\sigma_k) = \texttt{AGGREGATE}(\phi_L(h_{\sigma_k}^t, h_{\tau_k}^t, \forall \tau_k \in \mathcal{L}(\sigma_k)))$$

$$m_U^{t+1}(\sigma_k) = \texttt{AGGREGATE}(\phi_U(h_{\sigma_k}^t, h_{\tau_k}^t, \forall \tau_k \in \mathcal{U}(\sigma_k)))$$

and a combining step

$$h_{\sigma_k}^{t+1} = \texttt{COMBINE}(h_{\sigma_k}^t, m_B^{t+1}(\sigma_k), m_C^{t+1}(\sigma_k), m_L^{t+1}(\sigma_k), m_U^{t+1}(\sigma_k)),$$

where $t$ is the layer index of the SNN model, $\phi_n$ is the transformation function with adjacency of type $n$ (We use one layer MLP), $h_{\sigma_k}^t$ is the representation of the simplex $\sigma_k$ at layer $t$ with $h_{\sigma_k}^0$ being the initial embedding. We write the final representations of all the simplices in the simplicial complex $\mathcal{X}$ obtained from such an SNN model with $L$ layers compactly as

$$\mathbf{Z} = \mathcal{E}_\theta(\mathcal{X}, \mathbf{H}),$$

where $\theta$ denotes the set of learnable parameters and $\mathbf{H}$ is the initial feature matrix of all the simplices in $\mathcal{X}$. Although the above SNN model is based on MPSN, different SNN variants can be derived from the above model by choosing different `AGGREGATE` and `COMBINE` operators and which neighborhoods to aggregate.

## 4   The proposed `TopoSRL` pipeline

The main aim of the proposed approach is to learn an SNN encoder function $\mathcal{E}_\theta(\cdot)$ to compute expressive representations for simplices in a simplicial complex $\mathcal{X}$ in a self-supervised manner. Towards this end, we first compute two augmented views of a simplicial complex $\mathcal{X}$, namely, $\mathcal{X}^{(1)}$ and $\mathcal{X}^{(2)}$ (as described later on Section 4.1). Then we learn the representations of the two simplicial complexes through the encoder $\mathcal{E}_\theta(\cdot)$ as $\mathbf{Z}^{(1)} = \mathcal{E}_\theta(\mathcal{X}^{(1)}, \mathbf{H}^{(1)})$ and $\mathbf{Z}^{(2)} = \mathcal{E}_\theta(\mathcal{X}^{(2)}, \mathbf{H}^{(2)})$, where $\mathbf{Z}^{(i)} = \{\mathbf{Z}_1^{(i)}, \mathbf{Z}_2^{(i)}, \ldots, \mathbf{Z}_K^{(i)}\}$ is the set embedding matrices and $\mathbf{Z}_k^{(i)} \in \mathbb{R}^{N_k^{(i)} \times D}$ being the $D$-dimensional embedding matrix of all the $k$-simplices in the simplicial complex $\mathcal{X}^{(i)}$ and $N_k^{(i)}$ being the total number of $k$-simplices in $\mathcal{X}^{(i)}$. We contrast these representations and learn $\mathcal{E}_\theta(\cdot)$ by optimizing the simplicial contrastive loss (developed in Section 4.2). An illustration of the `TopoSRL` pipeline is shown in Figure 1(a).

### 4.1   Simplicial augmentation

We introduce a novel stochastic augmentation technique for simplicial complexes to generate two simplicial complexes $\mathcal{X}^{(1)}$ and $\mathcal{X}^{(2)}$ from $\mathcal{X}$ by adding open simplices and removing closed simplices at random via Bernoulli sampling. To begin with, we compute[1] all the open simplices in $\mathcal{X}$. Next,

---

[1]Identifying open simplices of order $k + 1$ in a simplicial complex $\mathcal{X}$ costs $\mathcal{O}(N_k^2)$, where $N_k$ is the total number of $k$-simplices. We only compute it once outside the training process.

---

**Algorithm 1** Simplicial complex augmentation

---

1: **procedure** SIMPLICIALAUGMENTATION($\mathcal{X}, \rho$)
2:     $\mathcal{X}_{\text{open}} \leftarrow \texttt{OpenSimplices}(\mathcal{X})$         $\triangleright$ Compute open $k$-simplices in $\mathcal{X}$ with $k \geq 2$
3:     **for** $i = 1$ and $2$ **do**
4:        $\mathcal{X}^{(i)} \leftarrow \mathcal{X}$
5:        **for** each $\sigma_k \in \mathcal{X}$ with $k > 1$ **do**
6:           Draw $r \sim \text{Bernoulli}(\rho)$
7:           **if** $r = 1$ **then**         $\triangleright$ remove a closed simplex
8:              $\mathcal{X}^{(i)} \leftarrow \{\mathcal{X}^{(i)} \setminus \sigma_k\}$
9:        **for** each $\sigma \in \mathcal{X}_{\text{open}}$ **do**
10:         Draw $a \sim \text{Bernoulli}(\rho)$
11:         **if** $a = 1$ **then**         $\triangleright$ add a open simplex
12:            $\mathcal{X}^{(i)} \leftarrow \{\mathcal{X}^{(i)} \cup \sigma\}$
13:     **return** $\mathcal{X}^{(1)}$ and $\mathcal{X}^{(2)}$

---

we draw a Bernoulli random variable for each open simplex to determine whether to include the open simplex in $\mathcal{X}^{(i)}$ for $i = 1, 2$. Similarly, we draw another Bernoulli random variable for each $k$-simplex ($k > 1$) in $\mathcal{X}$ to determine whether to retain the closed simplex in $\mathcal{X}^{(i)}$ for $i = 1, 2$. This procedure is summarized as Algorithm 1.

The proposed augmentation method preserves the inherent structure and relationships within the simplicial complex as illustrated in Figure 1(b). For example, consider the `walmart-trips` simplicial complex dataset[2], wherein a set of items purchased together by a customer in one trip to Walmart is a simplex. Now, if a customer purchases {peanut butter and jam}, {bread and jam}, and {peanut butter and bread} on three separate occasions, it is highly probable that the customer will buy {peanut butter, jam, bread} together on a subsequent trip. Adding the simplex {peanut butter, jam, bread} as in view 2 of Figure 1(b) will represent an instance of the simplicial complex if the data were sampled at a later point in time, and hence it still retains and provides better information to contrast compared to adding random simplices.

In contrast, in a random sampling method, where we add simplices at random, we need to include all the subsets of the simplex that is to be added, which costs $\mathcal{O}(2^k)$ for each $k$-order simplex for $k \in \{2, \ldots K\}$. As opposed to this, in the proposed simplicial augmentation technique, since we are adding an open simplex, all the subsets of this simplex are present in the simplicial complex, making the additional cost only $\mathcal{O}(1)$ and thereby making it computationally more efficient.

## 4.2 Simplicial contrastive loss

The proposed method aims to learn simplex representations by capturing topological information in higher-order simplices and their distant neighbors. In contrast, most of the existing self-supervised graph representation learning approaches primarily focus on contrasting corresponding nodes in augmented graphs and do not preserve topological information as the objective function does not focus on the relational distance between pairs of two nodes that are not neighbors.

**Cost matrices.** We define the following cost matrices to compute the `TopoSRL` loss function: the *intra-view cost matrix*, denoted by $\mathbf{C}_k^{(i)} \in \mathbf{R}^{N_k^{(i)} \times N_k^{(i)}}$ for $i = 1, 2$, measures the distance between representations of two simplices in the simplicial complex $\mathcal{X}^{(i)}$ and the *inter-view cost matrix*, denoted by $\mathbf{C}_k^{(1,2)} \in \mathbf{R}^{N_k^{(1)} \times N_k^{(2)}}$, measures the distance between the $k$-simplex in the two views. The intra-view cost matrix is used to minimize the relational distance between pairs of simplices from two augmented simplicial complexes. Specifically, the $(p, q)$th entry of $\mathbf{C}_k^{(i)}$, is defined as $[\mathbf{C}_k^{(i)}]_{p,q} = \|[\mathbf{Z}_k^{(i)}]_p - [\mathbf{Z}_k^{(i)}]_q\|_2^2$, where $[\mathbf{Z}_k^{(i)}]_p$ is the representation of the $k$-simplex $p$ in $\mathcal{X}^{(i)}$. On the other hand, the inter-view cost matrix is used to minimize the distance between the representation of a simplex and an aggregate representation of a sub-simplicial complex surrounding

---

[2]`https://www.cs.cornell.edu/~arb/data/walmart-trips/`

this simplex in the other augmented simplicial complex. The $(p, q)$th entry of $\mathbf{C}_k^{(1,2)}$ is defined as $[\mathbf{C}_k^{(1,2)}]_{p,q} = \|[\mathbf{Z}_k^{(1)}]_p - [\mathbf{Z}_k^{(2)}]_q\|_2^2$.

**Weight matrix.** We also construct a weight matrix $\mathbf{W}_k \in \mathbf{R}^{N_k^{(1)} \times N_k^{(2)}}$ for each $k$-simplex to capture the relation of a simplex $\sigma_k^{(1)} \in \mathcal{X}^{(1)}$ and $\sigma_k^{(2)} \in \mathcal{X}^{(2)}$, which will be used to calculate the aggregate representation of sub-simplicial complexes by assigning different importance weights that depend on how many hops $\sigma_k^{(2)}$ is away from $\sigma_k^{(1)}$ in the simplicial complex $\mathcal{X}^{(1)}$. We assign the entries of $\mathbf{W}_k$ as follows. If $\sigma_k^{(2)} = \sigma_k^{(1)}$, we assign a higher value $\eta_0$ to $[\mathbf{W}_k]_{\sigma_k^{(1)}, \sigma_k^{(2)}}$. If $\sigma_k^{(2)} \in \mathcal{L}(\sigma^{(1)}) \cup \mathcal{U}(\sigma^{(1)})$, i.e., if $\sigma_k^{(2)}$ is in one-hop neighborhood of $\sigma_k^{(1)}$ in $\mathcal{X}^{(1)}$, then we assign it a lower value $\eta_1$. Similarly, if $\sigma_k^{(2)}$ is in the two-hop neighborhood of $\sigma_k^{(1)}$ in $\mathcal{X}^{(1)}$, we assign a value of $\eta_2$, such that $\eta_0 > \eta_1 > \eta_2$ and then we apply row-wise softmax to $\mathbf{W}_k$ to normalize it. By construction, $\mathbf{W}_k$ is a row-stochastic matrix.

**Loss function.** The proposed simplicial contrastive loss is a convex combination of two terms, namely, the sub-simplicial complex loss $\mathcal{L}_{\text{sub}}$ and the relative simplicial complex loss $\mathcal{L}_{\text{rel}}$. The term $\mathcal{L}_{\text{sub}}$ measures the cumulative (over all the simplicies) weighted distance between the representation of a $k$-simplex $\sigma_k^i \in \mathcal{X}^{(1)}$ and $k$-simplices $\sigma_k^{i'} \in \mathcal{X}^{(2)}$ with appropriate weights in the matrix $\mathbf{W}_k$, and is given by

$$\mathcal{L}_{\text{sub}} = \sum_{k=0}^{K} \sum_{i=1}^{N_k^{(1)}} \sum_{j=1}^{N_k^{(2)}} [\mathbf{C}_k^{(1,2)}]_{i,j} [\mathbf{W}_k]_{i,j}. \tag{1}$$

The term $\mathcal{L}_{\text{rel}}$ calculates the relational distance between pairs of simplices and is given by

$$\mathcal{L}_{\text{rel}} = \sum_{k=0}^{K} \sum_{i,j=1}^{N_k^{(1)}} \sum_{i',j'=1}^{N_k^{(2)}} \left( [\mathbf{C}_k^{(1)}]_{i,j} - [\mathbf{C}_k^{(2)}]_{i',j'} \right)^2 [\mathbf{W}_k]_{i,i'} [\mathbf{W}_k]_{j,j'}. \tag{2}$$

For instance, consider two pairs of simplices $(\sigma_k^i, \sigma_k^j)$ and $(\sigma_k^{i'}, \sigma_k^{j'})$ such that $\sigma_k^i, \sigma_k^j \in \mathcal{X}_k^{(1)}$, $\sigma_k^{i'}, \sigma_k^{j'} \in \mathcal{X}_k^{(2)}$, $[\mathbf{W}_k]_{\sigma_k^i, \sigma_k^{i'}} > 0$, and $[\mathbf{W}_k]_{\sigma_k^j, \sigma_k^{j'}} > 0$, that is, $(\sigma_k^i, \sigma_k^{i'})$ and $(\sigma_k^j, \sigma_k^{j'})$ are distant neighbors (not necessarily one-hop). Hence, minimizing $\mathcal{L}_{\text{rel}}$ minimizes the squared difference of the distance between the representations of $(\sigma_k^i, \sigma_k^j)$ and $(\sigma_k^{i'}, \sigma_k^{j'})$, which leads to similar pairs of simplices from two distinct simplicial complexes maintaining an equal distance. In other words, if $\sigma_k^j$ is $m$-hop away from $\sigma_k^i$ and $\sigma_k^{j'}$ is $m$-hop away from $\sigma_k^{i'}$, then minimizing $\mathcal{L}_{\text{rel}}$ will reduce the difference between the distance of the pairs and preserve the $m$-hop information present in $\mathcal{X}$. Finally, the overall loss is

$$\mathcal{L} = \alpha \mathcal{L}_{\text{sub}} + (1 - \alpha) \mathcal{L}_{\text{rel}},$$

where $\alpha \in [0, 1]$ is a tunable parameter. The overall loss preserves the topological properties as adjacent simplices will be embedded more closely in the Euclidean space because of the $\mathcal{L}_{\text{sub}}$ term, capturing the local information. The difference in distance of contrasting simplex pairs will be minimized by minimizing $\mathcal{L}_{\text{rel}}$, capturing the global information.

We end this section with two interesting theoretical results[3] about the simplicial contrastive loss function and an overview of the TopoSRL pipeline in Algorithm 2. In the next proposition, we show that $\mathcal{L}_{\text{sub}}$ preserves the local information in a simplicial complex.

**Proposition 1.** *Minimizing $\mathcal{L}_{\text{sub}}$ is equivalent to jointly minimizing a lower bound on the distance between the representation of a simplex $\sigma_k^i \in \mathcal{X}^{(1)}$ and the aggregate representation of $\sigma_k^{i'} \in \mathcal{X}^{(2)}$ that is adjacent to $\sigma_k^i$ and the distance between the representation of $\sigma_k^{i'} \in \mathcal{X}^{(2)}$ and the aggregate of representations of $\sigma_k^{i'} \in \mathcal{X}^{(2)}$ that is adjacent to $\sigma_k^i \in \mathcal{X}^{(1)}$.*

Most graph SLL methods maximize MI between augmented graphs using MI estimators [9]. In the next theorem, we show that optimizing the proposed loss function is also related to optimizing MI between representations of simplices in one augmented simplicial complex and representations of

---

[3]Proofs are available in the supplementary material.

**Algorithm 2** TopoSRL

---

1:  **procedure** TOPOSRL($\mathcal{X}, \rho, \kappa, \mathbf{H}$)          ▷ $\kappa$ is the maximum order of the simplices of interest
2:      Initialize $\theta$
3:      $e \leftarrow 1$
4:      **while** $e \leq \#epochs$ **do**
5:          $\mathcal{X}^{(1)}, \mathcal{X}^{(2)} \leftarrow \text{SimplicialAugmentation}(\mathcal{X}, \rho)$
6:          Calculate $\mathbf{W}_k$ for $k = 0, \ldots, \kappa$
7:          $\mathbf{Z}^{(1)} \leftarrow \mathcal{E}_\theta(\mathcal{X}^{(1)}, \mathbf{H}^{(1)})$ and $\mathbf{Z}^{(2)} \leftarrow \mathcal{E}_\theta(\mathcal{X}^{(2)}, \mathbf{H}^{(2)})$.
8:          Calculate $\mathbf{C}_k^{(1)}, \mathbf{C}_k^{(2)}, \mathbf{C}_k^{(12)}$ for $k = 0, \cdots, \kappa$
9:          Minimize $\mathcal{L} = \alpha \mathcal{L}_{\text{sub}} + (1 - \alpha) \mathcal{L}_{\text{rel}}$
10:         $e = e + 1$
11:     $\mathbf{Z} \leftarrow \mathcal{E}_\theta(\mathcal{X}, \mathbf{H})$
12:     **return** $\mathbf{Z}$

---

its neighbors in the other augmented simplicial complex and representations of simplices and their neighborhoods within the same augmented simplicial complex conditioned on the input data.

Suppose $X$ represents the random variable corresponding to the input data, $X_k$ represents the $k$-simplices in $X$, and $X^{(i)}$ represents an augmented view of $X$, sharing the same sample space as $X$, and the input simplicial complex data $\mathcal{X} \sim X$. TopoSRL is designed to learn representations, denoted as $\mathbf{Z}$ for the input data and $\mathbf{Z}^{(i)}$ for its augmentation. We also denote $H(A)$ and $I(A, B)$ as the entropy of the random variable $A$ and the MI between the random variables $A$ and $B$, respectively.

**Theorem 1.** *Minimizing the expected loss $\mathcal{L}_{\text{sub}}$ (expectation is with respect to the random variable $X$) is equivalent to maximizing the MI between $\mathbf{Z}_k^{(i)}$ and $X_k$, i.e.,*

$$\underset{\theta}{\text{minimize}} \quad \mathcal{L}_{\text{sub},k} \equiv \underset{\theta}{\text{maximize}} \quad I(\mathbf{Z}_k^{(i)}, X_k), \tag{3}$$

*for $k = 0, 1, \ldots, K$, where $\mathcal{L}_{\text{sub},k}$ is the $k$th summand in (1), $I(\mathbf{Z}_k^{(i)}, X_k)$ is the MI between the representations of the augmented $k$-simplices $\mathbf{Z}_k^{(i)}$ and the $k$-simplicies in the originial data $X_k$.*

The above theorem shows that in expectation, minimizing the term $\mathcal{L}_{\text{sub}}$ is equivalent to minimizing the variance between representations in the augmented simplicial complexes, which leads to minimization of the conditional entropy $H(\mathbf{Z}_k^{(i)}|X_K)$, implying the maximization of MI as $I(\mathbf{Z}_k^{(i)}, X_k) = H(\mathbf{Z}_k^{(i)}) - H(\mathbf{Z}_k^{(i)}|X_k)$. The proof also shows that minimizing the simplicial contrastive loss implicitly maximizes MI between simplices from one augmented simplicial complex and representations of its neighbors in the other augmented simplicial complex conditioned on the input and representations of simplices and their neighborhoods within the same augmented simplicial complex. This result is along the lines of earlier methods like DGI [9] and InfoGraph [22]. However, TopoSRL does not require additional components in DGI or InfoGraph for MI maximization.

## 5  Experiments

We follow the standard setting of self-supervised learning methods. Firstly, we train the encoder with the proposed simplicial contrastive loss. Next, we freeze the encoders' model parameters, extract representations for all the simplices, and train a classifier for the following two downstream tasks. The code is available at `https://github.com/HirenMadhu/TopoSRL`.

**Downstream tasks.** We focus on *node classification*, *simplicial closure*, *trajectory prediction* and *graph classification* tasks. Node classification focuses on predicting labels for 0-simplices (aka nodes) in a given simplicial complex. We perform node classification on the following publicly available datasets[4], namely, `contact-primary-school`, `contact-high-school`, `senate-bills`. We use classification accuracy as the performance metric. In simplicial closure, the aim is to predict the closure of open simplices in a time series of simplicial complex data. We perform simplicial

---

[4]Dataset details are presented in the supplementary material. Datasets are available at `https://www.cs.cornell.edu/~arb/data/`

| Method | Type | high-school | primary-school | senate-bills |
|---|---|---|---|---|
| GCN | S | 0.4±0.04 | 0.30±0.04 | 0.67±0.06 |
| GraphSage | S | 0.27±0.05 | 0.37±0.05 | 0.54±0.03 |
| GIN | S | 0.18±0.04 | 0.16±0.02 | 0.53±0.04 |
| GAT | S | 0.34±0.05 | 0.19±0.06 | 0.5±0.04 |
| CCA-SSG | SSL | 0.68±0.16 | 0.14±0.07 | 0.62±0.04 |
| GCA | SSL | 0.18±0.08 | 0.12±0.05 | 0.5±0.0 |
| BGRL | SSL | 0.11±0.01 | 0.09±0.01 | 0.5±0.0 |
| GraphMAE | SSL | 0.78±0.05 | 0.2±0.02 | 0.57±0.01 |
| SAN | S | 0.86± 0.04 | 0.29± 0.06 | 0.53 ± 0.09 |
| SCNN | S | 0.81 ± 0.01 | 0.67±0.04 | 0.615 ± 0.05 |
| MPSN | S | 0.89 ± 0.01 | **0.79 ± 0.06** | **0.75 ± 0.05** |
| TopoSRL | SSL | **0.92 ± 0.05** | 0.61 ± 0.05 | 0.72± 0.06 |

Table 1: Node classification accuracies on simplicial complex datasets; S stands for supervised setting, and SSL stands for self-supervised setting. (Best accuracy is bold and second best accuracy is underline)

closure on `email-Eu`, `email-Enron`, `contact-high-school` datasets. We first split the data across time on these temporal datasets and then train the encoder on the first 80% of the data. The last 20% is used for inference. Since the class distribution is very skewed, we use F1-macro to evaluate the performance. Trajectory prediction focuses on predicting the next node in a trajectory, given a sequence of nodes. We perform experiments on two datasets, namely, `Ocean` and the `Synthetic` dataset generated using the method described in [6]. We use accuracy as the metric for comparison. In graph classification, the focus is on learning representations for the whole graph and classifying them. Clique lifting is used to convert a graph to a simplicial complex and then use an SNN model to extract representations for the whole graph. We evaluate `TopoSRL`'s performance in graph classification task on `PROTEINS`, `NCI1`, `REDDIT-B`, `REDDIT-M` and `IMDB-B` datasets from the TUDatasets [23] repository.

**Baselines.** We compare TopoSRL against several supervised techniques employing architectures such as SAN [1], SCNN [3], and MPSN [5]. This comparison builds confidence in the expressive capabilities of `TopoSRL` and solidifies its usefulness in learning simplex representations in settings with less-labeled or unlabeled data. Additionally, we conduct experiments involving various graph neural network models, including graph convolutional network (GCN) [24], GAT[15], GIN [25], and GraphSage[26]. Furthermore, we also compare our method with the current state-of-the-art graph SSL methods CCA-SSG[13], BGRL [11], GraphMAE [14] and GCA [7]. This comparison also clarifies that in the SSL setting, the proposed simplicial SSL method (i.e., `TopoSRL`) performs better than graph SSL methods, as demonstrated by the existing supervised simplicial representation learning methods. For `TopoSRL`, we use a 3-layer MPSN as the encoder network $\mathcal{E}_\theta(\cdot)$. Details about the experimental setup, hyperparameters, and results with other SNN encoder models are available in the supplementary material. We follow the standard practice where all the results are averaged over ten different seeds, and one run is performed for each seed.

**Results and discussion.** Our results demonstrate that `TopoSRL` consistently outperforms state-of-the-art graph SSL methods. In Table 1, we see that `TopoSRL` surpasses supervised methods on graphs and state-of-the-art graph SSL techniques while being competitive with supervised approaches for the node classification task. `TopoSRL` has a superior performance compared to MPSN on the `contact-high-school` dataset while being competitive with MPSN on the other two datasets. As can be seen in experiments reported in the supplementary material, `contact-primary-school`, being a denser dataset, does not benefit significantly from minimizing both the terms, but minimizing only $\mathcal{L}_{\text{rel}}$ (i.e., $\alpha = 1$) improves performance.

Table 2 showcases experimental results on simplicial closure datasets. On this task, we see that the less parameterized SCNN model outperforms both `TopoSRL` and MPSN on the `email-Enron` and `contact-primary-school` datasets, while `TopoSRL` performs better than MPSN and is competitive overall. These results can be attributed to the skewed nature of simplicial closure datasets, where models with large parameters like MPSN and SAN are prone to overfitting labels. Although TopoSRL

| Method | Type | email-Enron | contact-primary-school | contact-high-school |
|--------|------|-------------|------------------------|---------------------|
| SAN | S | 0.57±0.12 | 0.39±0.03 | 0.3±0.07 |
| SCNN | S | **0.61±0.08** | **0.49±0.09** | 0.41±0.11 |
| MPSN | S | 0.43±0.07 | 0.43±0.05 | **0.47±0.20** |
| TopoSRL | SSL | 0.59±0.05 | 0.46±0.01 | 0.43±0.0 |

Table 2: Simplicial closure performance using F1 scores; S stands for supervised setting and SSL stands for self-supervised setting.

| Method | Type | Ocean | Synthetic |
|--------|------|-------|-----------|
| Projection | S | 27.15±0.0 | 52.0±0.0 |
| SCoNe | S | 30.0±0.6 | **55.4±1.1** |
| SCNN | S | 28.5±0.6 | 50.5±1.0 |
| TopoSRL | SSL | **42.0±3.0** | 50.0±1.0 |

Table 3: Trajectory prediction performance using accuracy scores; S stands for supervised setting, and SSL stands for self-supervised setting.

focuses on preserving topology rather than concentrating on a specific downstream task, it has improved performance compared to these models.

Table 3 presents the results on trajectory prediction. As we can see, the results indicate that TopoSRL outperforms SCNN and ScoNe when applied to the Ocean dataset. When tested on the synthetic dataset, TopoSRL demonstrates performance on par with a supervised SCNN. These results further highlight the expressive representation capabilities of TopoSRL on oriented simplicial complexes and its use cases in practical applications. As we can see in Table 4, TopoSRL performs on par with supervised graph baselines and simplicial baselines. Further, TopoSRL outperforms or performs on par with graph SSL baselines, showing the advantages of TopoSRL on a standard graph dataset with clique lifting compared to graph SSL methods.

| Method | Type | PROTEINS | NCI1 | REDDIT-B | REDDIT-M | IMDB-B |
|--------|------|----------|------|----------|----------|--------|
| GCN | S | 0.58 | 0.53 | 0.71 | 0.49 | 0.69 |
| GraphSage | S | 0.61 | 0.48 | 0.69 | 0.5 | 0.68 |
| GIN | S | 0.61 | **0.702** | **0.73** | 0.57 | 0.71 |
| GAT | S | 0.57 | 0.45 | 0.53 | 0.5 | 0.6 |
| CCA-SSG | SSL | 0.64 | 0.67 | 0.72 | 0.58 | 0.675 |
| BGRL | SSL | 0.62 | 0.63 | 0.72 | **0.61** | 0.66 |
| SAN | S | 0.64 | **0.702** | 0.71 | 0.59 | **0.72** |
| SCNN | S | 0.61 | 0.370 | 0.69 | 0.57 | 0.67 |
| MPSN | S | 0.63 | **0.702** | 0.71 | 0.59 | 0.67 |
| TopoSRL | SSL | **0.75** | 0.700 | 0.72 | 0.606 | 0.695 |

Table 4: Graph classification accuracies on TUDatasets; S stands for supervised setting, and SSL stands for self-supervised setting.

To further validate the efficacy of our augmentation technique, we conduct an ablation study, comparing TopoSRL with and without the proposed augmentation strategy. The results indicate a significant performance improvement when incorporating the proposed simplicial augmentation, underlining its crucial role in capturing long-range dependencies and higher-order interactions. Table 5 shows that our augmentation technique results in considerable performance gains over random augmentation methods. We observe that only adding open simplices and not removing closed simplices is a better augmentation technique than random augmentation, but its performance deteriorates compared to the proposed augmentation technique. This occurs because the contrastive objective function is more effective when there is more information to contrast. Removing closed simplices allows the contrastive loss and encoder to contrast more information, resulting in higher performance instead of only adding open simplices. Additionally, we have observed that incorporating higher-order information from simplices with more than three vertices leads to reduced performance. This reduction is due to the limited presence of higher-order simplices in these datasets, which causes overparameterization of

| Dataset | Order | R | OO | O |
|---|---|---|---|---|
| contact-high-school | 3 | 0.62±0.05 | 0.81±0.04 | **0.92±0.05** |
| contact-high-school | 4 | 0.6±0.05 | 0.78±0.03 | **0.85±0.06** |
| contact-primary-school | 3 | 0.36±0.06 | 0.51±0.04 | **0.61±0.05** |
| contact-primary-school | 4 | 0.4±0.09 | 0.44±0.08 | **0.52±0.06** |
| senate-bills | 3 | 0.52±0.07 | 0.66±0.05 | **0.72±0.06** |

Table 5: Performace on node classification using random augmentation; R stands for random augmentation, OO stands for adding only open simplices augmentation, and O stands for augmentation with open simplicies using Algorithm 1.

the model and a slight decline in performance. More experimental results comparing different SNN encoders and different values of alpha are reported in the supplementary material.

To test the ability of TopoSRL to learn expressive representations, we perform the following experiment: 1) Use the pre-trained TopoSRL encoders to extract representations and 2) Use only a partially labeled (e.g., 20% train and 80% test, 40% train and 60% test, etc.) data to train a logistic regression classifier for node classification task in the contact-high-school dataset. MPSN and GCN are trained with cross-entropy loss on the train set. Since the weights for the TopoSRL encoder trained without labels have been saved, no new encoders were trained to produce the results. As we can see in Table 6, with an increase in the size of the train set, the performance increases across all the methods. Furthermore, TopoSRL has a significantly improved performance of about 5% over supervised MPSN in the 20-80 and 40-60 split. This provides empirical evidence about the expressive capabilities of TopoSRL and its efficacy with less-labeled data. Hence, TopoSRL would be preferable over standard supervised models in the less-labeled data setting.

| Method | Type | 20-80 | 40-60 | 60-40 | 80-20 |
|---|---|---|---|---|---|
| GCN | S | 0.31±0.04 | 0.34±0.03 | 0.36±0.04 | 0.4±0.04 |
| CCA-SSG | SSL | 0.39±0.04 | 0.45±0.04 | 0.53±0.04 | 0.68±0.16 |
| BGRL | SSL | 0.48±0.00 | 0.49±0.00 | 0.48±0.00 | 0.51±0.00 |
| SCNN | S | 0.71±0.02 | 0.74±0.02 | 0.81±0.04 | 0.86±0.04 |
| MPSN | S | 0.74±0.02 | 0.80±0.02 | **0.86±0.01** | 0.89±0.01 |
| TopoSRL | SSL | **0.79±0.02** | **0.84±0.01** | 0.86±0.02 | **0.92±0.05** |

Table 6: Node classification accuracies on contact-high-school; S stands for supervised setting, and SSL stands for self-supervised setting, 20-80 refers to 20% data for training, 80 percent for test.

## 6   Conclusions

We have introduced a novel framework for SSL on simplicial complexes by leveraging topological properties underlying a simplicial complex. In particular, we have proposed two key components: a stochastic simplicial augmentation method and a simplicial contrastive loss function, which collectively provide a mechanism for learning representations that retain local and global topological information in the simplicial complex. The proposed simplicial augmentation method offers the advantage of generating topologically consistent views of the original simplicial complex, allowing the model to learn a rich set of features efficiently. The contrastive loss function comprises two primary terms: the sub-simplicial complex loss and the relative simplicial complex loss. The former focuses on minimizing the distance between the representation of a simplex and its adjacent simplices in the augmented complex, thereby preserving local information. The latter aims to maintain relative spatial relationships present in the original simplicial complex, thereby capturing global information. We have also theoretically proved that the proposed loss function is related to the MI objective function, which is commonly used in SSL on graphs. Our model outperforms state-of-the-art graphs SSL methods on a variety of datasets. We believe this work lays a solid foundation for further exploration of SSL methods for simplicial complexes, e.g., via alternative augmentation methods, contrastive loss functions, scalable models, and new applications, potentially opening up new avenues for topological data analysis.

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
