# Supplementary material for
# `TopoSRL`: Topology Preserving Self-Supervised Simplicial Representation Learning

**Hiren Madhu and Sundeep Prabhakar Chepuri**
Indian Insitute of Science
`hirenmadhu, spchepuri@iisc.ac.in`

## 1  Proof of Proposition 1 and Theorem 1 from the paper

**Proposition 1.** *Minimizing $\mathcal{L}_{\mathrm{sub}}$ is equivalent to jointly minimizing a lower bound on the distance between the representation of a simplex $\sigma_k^i \in \mathcal{X}^{(1)}$ and the aggregate representation of $\sigma_k^{i'} \in \mathcal{X}^{(2)}$ that is adjacent to $\sigma_k^i$ and the distance between the representation of $\sigma_k^{i'} \in \mathcal{X}^{(2)}$ and the aggregate of representations of $\sigma_k^{i'} \in \mathcal{X}^{(2)}$ that is adjacent to $\sigma_k^i$.*

*Proof.* Assume that $\mathcal{X}^{(1)}$ and $\mathcal{X}^{(2)}$ are two augmented simplicial complexes obtained from the given simplicial complex $\mathcal{X}$ using the simplicial complex augmentation method described in Algorithm 1. We can express $\mathcal{L}_{\mathrm{sub},k}$, the $k$th summand in Equation 1 in main paper, as

$$
\mathcal{L}_{\mathrm{sub},k} = \sum_{i=1}^{N_k^{(1)}} \sum_{j=1}^{N_k^{(2)}} [\mathbf{C}_k^{(1,2)}]_{i,j}[\mathbf{W}_k]_{i,j} = \sum_{i=1}^{N_k^{(1)}} \sum_{j=1}^{N_k^{(2)}} [\mathbf{W}_k]_{i,j} \| [\mathbf{Z}_k^{(1)}]_i - [\mathbf{Z}_k^{(2)}]_j \|_2^2
$$
$$
= \sum_{i=1}^{N_k^{(1)}} \sum_{i' \in \mathcal{N}_k^{(1)}(i)} [\mathbf{W}_k]_{i,i'} \| [\mathbf{Z}_k^{(1)}]_i - [\mathbf{Z}_k^{(2)}]_{i'} \|_2^2, \qquad \text{(S1)}
$$

where the set $\mathcal{N}_k^{(1)}(i) = \{l \,|\, [\mathbf{W}_k]_{i,l} > 0\}$ collects the $k$-simplices in $\mathcal{X}^{(2)}$ surrounding the $i$th $k$-simplex in $\mathcal{X}^{(1)}$ as described by the weight matrix. The simplices not in $\mathcal{N}_k^{(1)}(i)$ do not contribute to the sum as the corresponding entries in $\mathbf{W}_k$ are zero.

Now, let us define $[\mathbf{Z}_k^{(2)}]_{\mathrm{agg}(i)} = \sum_{i' \in \mathcal{N}_k^{(1)}(i)} [\mathbf{W}_k]_{i,i'}[\mathbf{Z}_k^{(2)}]_{i'}$ as the aggregate representation of the $k$-simplices in $\mathcal{N}_k^{(1)}(i)$. Then we have

$$
\mathcal{L}_{\mathrm{sub},k} = \sum_{i=1}^{N_k^{(1)}} \sum_{i' \in \mathcal{N}_k^{(1)}(i)} [\mathbf{W}_k]_{i,i'} \| ([\mathbf{Z}_k^{(1)}]_i - [\mathbf{Z}_k^{(2)}]_{\mathrm{agg}(i)}) + ([\mathbf{Z}_k^{(2)}]_{\mathrm{agg}(i)} - [\mathbf{Z}_k^{(2)}]_{i'}) \|_2^2
$$
$$
\overset{(a)}{\leq} \sum_{i=1}^{N_k^{(1)}} \sum_{i' \in \mathcal{N}_k^{(1)}(i)} [\mathbf{W}_k]_{i,i'} \| [\mathbf{Z}_k^{(1)}]_i - [\mathbf{Z}_k^{(2)}]_{\mathrm{agg}(i)} \|_2^2 + [\mathbf{W}_k]_{i,i'} \| [\mathbf{Z}_k^{(2)}]_{\mathrm{agg}(i)} - [\mathbf{Z}_k^{(2)}]_{i'} \|_2^2
$$
$$
\overset{(b)}{=} \sum_{i=1}^{N_k^{(1)}} \| [\mathbf{Z}_k^{(1)}]_i - [\mathbf{Z}_k^{(2)}]_{\mathrm{agg}(i)} \|_2^2 + \sum_{i=1}^{N_k^{(1)}} \sum_{i' \in \mathcal{N}_k^{(1)}(i)} [\mathbf{W}_k]_{i,i'} \| [\mathbf{Z}_k^{(2)}]_{\mathrm{agg}(i)} - [\mathbf{Z}_k^{(2)}]_{i'} \|_2^2 \quad \text{(S2)}
$$

where (a) is due to the triangle inequality and (b) is due to row stochasticity of $\mathbf{W}_k$. $\qquad\square$

37th Conference on Neural Information Processing Systems (NeurIPS 2023).

**Theorem 1.** *Minimizing the expected loss $\mathcal{L}_{\text{sub}}$ (expectation is with respect to the random variable $X$) is equivalent to maximizing the MI between $\mathbf{Z}_k^{(i)}$ and $X_k$, i.e.,*

$$\underset{\theta}{\text{minimize}} \quad \mathcal{L}_{\text{sub},k} \equiv \underset{\theta}{\text{maximize}} \quad I(\mathbf{Z}_k^{(i)}, X_k), \tag{S3}$$

*for $k = 0, 1, \ldots, K$, where $\mathcal{L}_{\text{sub},k}$ is the kth summand in* (S1), *$I(\mathbf{Z}_k^{(i)}, X_k)$ is the MI between the representations of the augmented k-simplices $\mathbf{Z}_k^{(i)}$ and the k-simplicies in the originial data $X_k$.*

*Proof.* Assume that $\mathcal{X}^{(1)} \sim p(\cdot|\mathcal{X})$ and $\mathcal{X}^{(2)} \sim p(\cdot|\mathcal{X})$. The two views come from a probability distribution conditioned on original data distribution $X$, and $\mathcal{X}$ is as distributed as $X$. The sample mean of $\sum_{i=1}^{N_k^{(1)}} [\mathbf{C}_k^{(1,2)}]_{i,j}[\mathbf{W}_k]_{i,j}$ can be approximated by the expectation with respect to the data distribution and the conditional augmentation distribution as

$$\frac{1}{N_k^{(1)}} \mathcal{L}_{\text{sub},k} \approx \mathbb{E}_{\mathcal{X}} \, \mathbb{E}_{\mathcal{X}^{(1)}, \mathcal{X}^{(2)} \sim p(\cdot|\mathcal{X})} \sum_{i' \in \mathcal{N}_k^{(1)}(i)} [\mathbf{C}_k^{(1,2)}]_{i,j}[\mathbf{W}_k]_{i,j}$$

$$\overset{(c)}{\leq} \mathbb{E}_{\mathcal{X}} \, \mathbb{E}_{\mathcal{X}^{(1)}, \mathcal{X}^{(2)} \sim p(\cdot|\mathcal{X})} [\mathbf{W}_k]_{i,i'} \| [\mathbf{Z}_k^{(1)}]_i - [\mathbf{Z}_k^{(2)}]_{\text{agg}(i)} \|_2^2$$

$$+ \sum_{i' \in \mathcal{N}_k^{(1)}(i)} [\mathbf{W}_k]_{i,i'} \| [\mathbf{Z}_k^{(2)}]_{\text{agg}(i)} - [\mathbf{Z}_k^{(2)}]_{i'} \|_2^2, \tag{S4}$$

where (c) follows from (a) in Equation (S2).

Suppose we have $T$-dimensional features. Then the first term simplifies to

$$\mathbb{E}_{\mathcal{X}} \, \mathbb{E}_{\mathcal{X}^{(1)}, \mathcal{X}^{(2)} \sim p(\cdot|\mathcal{X})} [\mathbf{W}_k]_{i,i'} \| [\mathbf{Z}_k^{(1)}]_i - [\mathbf{Z}_k^{(2)}]_{\text{agg}(i)} \|_2^2$$

$$= \mathbb{E}_{\mathcal{X}} \, \mathbb{E}_{\mathcal{X}^{(1)}, \mathcal{X}^{(2)} \sim p(\cdot|\mathcal{X})} \sum_{d=1}^{T} [\mathbf{W}_k]_{i,i'} ([\mathbf{Z}_k^{(1)}]_{i,d} - [\mathbf{Z}_k^{(2)}]_{\text{agg}(i),d})^2$$

$$= [\mathbf{W}_k]_{i,i'} \mathbb{E}_{\mathcal{X}} \, \mathbb{E}_{\mathcal{X}^{(1)}, \mathcal{X}^{(2)} \sim p(\cdot|\mathcal{X})} \sum_{d=1}^{T} [\mathbf{Z}_k^{(1)}]_{i,d}^2 + [\mathbf{Z}_k^{(2)}]_{\text{agg}(i),d}^2 - 2[\mathbf{Z}_k^{(1)}]_{i,d}[\mathbf{Z}_k^{(2)}]_{\text{agg}(i),d}$$

$$= [\mathbf{W}_k]_{i,i'} [\mathbb{E}_{\mathcal{X}} [\sum_{d=1}^{T} \mathbb{E}_{\mathcal{X}^{(1)} \sim p(\cdot|\mathcal{X})} [[\mathbf{Z}_k^{(1)}]_{i,d}^2] + \mathbb{E}_{\mathcal{X}^{(2)} \sim p(\cdot|\mathcal{X})} [[\mathbf{Z}_k^{(2)}]_{\text{agg}(i),d}^2]$$

$$- 2\mathbb{E}_{\mathcal{X}^{(1)} \sim p(\cdot|\mathcal{X})} [[\mathbf{Z}_k^{(1)}]_{i,d} \mathbb{E}_{\mathcal{X}^{(2)} \sim p(\cdot|\mathcal{X})} [[\mathbf{Z}_k^{(2)}]_{\text{agg}(i),d}]]. \tag{S5}$$

Since $\mathcal{X}^{(1)}$ and $\mathcal{X}^{(2)}$ are independently drawn from an identical distribution, the expectations of their encoded features are the same. Hence we have

$$= \mathbb{E}_{\mathcal{X}} \sum_{d=1}^{D} 2\mathbb{E}_{\mathcal{X}^{(1)} \sim p(\cdot|\mathcal{X})} [[\mathbf{Z}_k^{(1)}]_{i,d}^2] - 2\mathbb{E}_{\mathcal{X}^{(1)} \sim p(\cdot|\mathcal{X})} [[\mathbf{Z}_k^{(1)}]_{i,d}]^2$$

$$= 2\mathbb{E}_{\mathcal{X}} \sum_{d=1}^{D} \mathbb{E}_{\mathcal{X}^{(1)} \sim p(\cdot|\mathcal{X})} [[\mathbf{Z}_k^{(1)}]_{i,d}^2] - \mathbb{E}_{\mathcal{X}^{(1)} \sim p(\cdot|\mathcal{X})} [[\mathbf{Z}_k^{(1)}]_{i,d}^2$$

$$= 2\mathbb{E}_{\mathcal{X}} \sum_{d=1}^{D} \text{variance}_{\mathcal{X}^{(1)} \sim p(\cdot|\mathcal{X})} ([\mathbf{Z}_k^{(1)}]_{i,d}). \tag{S6}$$

This suggests that minimizing $\mathcal{L}_{\text{sub},k}$ reduces the variance of the representations of simplices from one augmented simplicial complex and representations of its neighbors in the other augmented simplicial complex conditioned on the input. A similar result can be established for the second term in Equation (S4), which will reduce the variance of representations of simplices and their neighborhoods within the same augmented simplicial complex.

It is known that $H(\mathbf{Z}_k^{(1)}|X_k) = \sum_d H([\mathbf{Z}_k^{(1)}]_d|X_k)$ when the entries of $\mathbf{Z}_k^{(1)}$ are independent. For the one-dimensional Gaussian distribution with variance $\sigma^2$, the entropy is equal to $\frac{1}{2} \log(2\pi \exp \sigma^2)$.

| Dataset | Simplex | #0-simplices | #1-simplices | #2-simplices |
|---|---|---|---|---|
| `contact-high-school` | Group of people | 327 | 5818 | 2370 |
| `contact-primary-school` | Group of people | 242 | 8317 | 5139 |
| `senate-bills` | Co-sponsors | 294 | 6974 | 3013 |
| `email-Enron` | Email groups | 142 | 1655 | 6095 |

Table S1: Dataset statistics.

In our case, $H(\mathbf{Z}_k^{(1)}|X_k) = \sum_d \frac{1}{2} \log(2\pi \exp(\text{variance}[[\mathbf{Z}_k^{(1)}]_d]))$. Consequently, minimizing the variance of features in each dimension reduces its entropy conditioned on the input. For mutual information (MI), we know that $I(\mathbf{Z}_k^{(1)}, X_k) = H(\mathbf{Z}_k^{(1)}) - H(\mathbf{Z}_k^{(1)}|X_k)$.

Therefore, by minimizing Equation 1 from the main paper, the MI $I(\mathbf{Z}_k^{(1)}, X_k)$ is maximized.  □

## 2 Details related to experiments

**Datasets.** In Table S1, we provide details about the datasets used in the experiments in the paper, namely, `contact-high-school`, `contact-primary-school`, `senate-bills`, and `email-Enron`. A simplex in `contact-high-school` and `contact-primary-school` represent a group of people who were in close proximity, and the classes are the classrooms that the students are in. In `senate-bills`, a simplex is the set of co-sponsors of bills that are put forth in the Senate, and the classes are the political party the sponsors belong to. In `email-Enron`, a simplex represents a set of users in an email group.

**Feature Initialization.** To initialize features for the simplices in both the simplicial complexes $\mathcal{X}^{(1)}$ and $\mathcal{X}^{(2)}$, we follow the procedure described next.

Every simplicial complex has an underlying graph, which is a collection of 1-simplices. Firstly, we determine the diameter $D$ of this underlying graph, where the diameter is defined as the shortest path length between the most distant nodes in the graph. Subsequently, choose $D$ random nodes from the graph and designate them as anchor nodes. If the mean distance between all the pairs of anchor nodes is not greater than a threshold $\epsilon$, repeat the above step. Next, we calculate the distance of each node from the anchor nodes, identifying the closest anchor node for all nodes. We assign a one-hot vector of the nearest anchor node as the features. Finally, to obtain the features of a simplex, we perform an elementwise `OR` operation on the features of all nodes within the simplex.

Anchor nodes serve as fixed reference points within a simplicial complex, anchoring its structure and providing stability. They can serve as important anchor points for capturing and encoding the underlying patterns and relationships in the simplicial complex. By ensuring that the anchor nodes are far enough from each other, we ensure that the initial features $\mathbf{H}$, represent distinct discriminative information about different parts of a simplicial complex. Furthermore, anchor nodes can also represent important entities. For example, in social networks, anchor nodes could be influential individuals or key opinion leaders. With this initialization method, a simplex can represent the group of influential individuals a user group follows, embedding the semantic information in the initial features.

**Setup.** To ensure a fair comparison, we use three (message passing) layers for all the neural models (i.e., SNNs and GNNs). The encoder parameters $\theta$ are optimized using the Adam optimizer [1] with a constant learning rate of $10^{-3}$ and a weight decay of $10^{-4}$ across all experiments. We train the encoder for 20 epochs on every dataset for the proposed model. We maintain a consistent approach for setting the feature dimensions across all datasets, setting the hidden dimensions output to 10 times the graph's diameter and the output layer's output dimension to 20 times the graph's diameter in a self-supervised setting or equal to the number of classes. In self-supervised experiments, we first encode the simplicial complex using the learned encoder. We subsequently train a logistic regression classifier or a single-layer MLP to obtain performance metrics. We use $\rho = 0.1$ for simplicial complex augmentation for all the datasets.

To construct the weight matrix $\mathbf{W}_k$, we use $\eta_0 = 5$, $\eta_1 = 3$, and $\eta_2 = 1$. Moreover, calculating the term $\mathcal{L}_{\text{rel}}$ for all the entries of $\mathbf{W}_k$ matrices is not needed (more discussion on this is provided

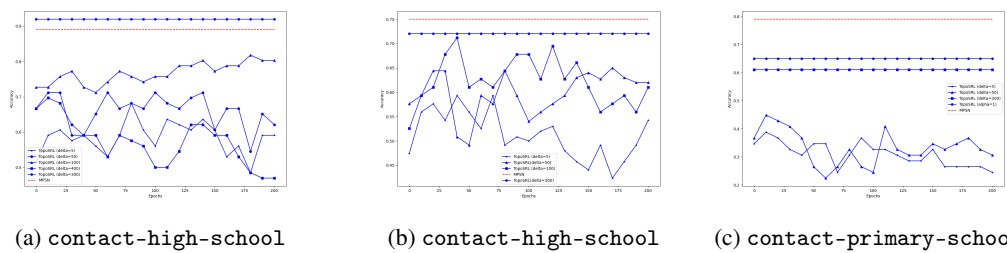

| (a) `contact-high-school` | (b) `contact-high-school` | (c) `contact-primary-school` |

Figure S1: Performance comparison plots for varying $\delta$ and number of epochs.

|  | contact-high-school | contact-primary-school | senate-bills |
|---|---|---|---|
| $\alpha = 1$ | 0.80±0.07 | **0.65±0.08** | 0.67±0.06 |
| $\alpha = 0.75$ | 0.87±0.04 | 0.61±0.1 | 0.64±0.05 |
| $\alpha = 0.5$ | **0.92±0.05** | 0.61±0.05 | **0.72±0.06** |
| $\alpha = 0.25$ | 0.8±0.05 | 0.57±0.08 | 0.66±0.04 |
| $\alpha = 0$ | 0.78±0.08 | 0.48±0.07 | 0.62±0.07 |

Table S2: Results with varying alpha on node classification datasets.

later on). Hence, to reduce training complexity, we fix the number of randomly sampled values $\delta$ for computing $\mathcal{L}_{\mathrm{rel}}$ to $\delta = 300$ for all encoders. The runtime of `TopoSRL` is depends upon the original simplicial complex $\mathcal{X}$ size, epoch number, and sampled indices $\delta$. Typically, our setup necessitates approximately 2 to 3 hours of training time for all datasets on a single NVIDIA Quadro RTX 8000.

# 3 Additional experimental results

As illustrated in Figures S1a, S1b, and S1c, we find a positive correlation between an increase in $\delta$ and the improvement in classification accuracy for datasets `contact-high-school` and `senate-bills`, but after it crosses a value of 300, the performance decreases significantly for the dataset `contact-high-school`. The results provided in the paper are with $\delta = 300$. This observation suggests that the choice of $\delta$ plays a crucial role in the performance of our model. Additionally, we notice an interesting pattern during training, where the performance reaches its peak at approximately 20 epochs. Beyond this point, the performance either experiences a decline or yields diminishing gains, indicating the possibility of overfitting or convergence. In light of these observations, we decided to train the encoder for 20 epochs.

Furthermore, we conducted additional experiments to analyze the impact of the parameter $\alpha$ in total loss on the model's performance. Specifically, we test the model with five distinct values of $\alpha$ to observe any variations in the results. These additional experiments reported in Table S2 provide valuable insights about the sensitivity of the model to the choice of $\alpha$. As can be seen, only minimizing the term $\mathcal{L}_{\mathrm{rel}}$ with $\alpha = 0$ does not offer any benefits. For `contact-primary-school`, focusing solely on minimizing the $\mathcal{L}_{\mathrm{sub}}$ term yields superior results compared to others minimizing both. However, it is worth noting that there exists a notable performance gap between the supervised method and our proposed `TopoSRL` approach for this particular dataset. In essence, simultaneous minimization of both terms in our objective function generally leads to superior outcomes on the `contact-high-school` and `senate-bills` datasets.

Furthermore, we also report the performance of `TopoSRL` with other SNN models for the encoder functions such as SAN[2] and SCNN[3] in Table S3. The SCNN model being a less expressive model, might not completely capture the inherent complexities and higher-order interactions in the `contact-high-school` and `contact-primary-school` datasets, leading to its lower performance. Conversely, the SAN model, which relies on attention mechanisms, might be overwhelmed by the high density of connections in `contact-primary-school`, causing its performance to suffer due to imbalances in the attention weights assigned to different nodes.

| Data | MPSN | SAN | SCNN |
|---|---|---|---|
| contact-high-school | 0.92±0.05 | **0.93±0.05** | 0.86±0.05 |
| contact-primary-school | **0.71±0.05** | 0.57±0.06 | 0.48±0.08 |
| senate-bills | **0.72±.06** | 0.64±0.06 | 0.71±0.06 |

Table S3: Node classification using different encoder models in TopoSRL.

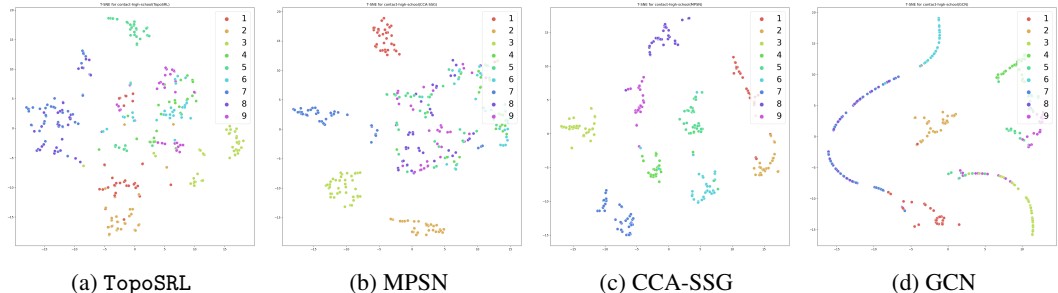

(a) TopoSRL     (b) MPSN     (c) CCA-SSG     (d) GCN

Figure S2: Comparison of TSNE plots of representations learned by various encoders.

Figure S2 compares T-SNE plots for four different methods: TopoSRL, CCA-SSG, MPSN, and GCN. The dataset used is contact-high-school simplicial complex, and the node classes are the prediction target. Each point in the plot corresponds to the representations of a node from the contact-high-school dataset. To restate what is mentioned in the manuscript, MPSN can cluster most nodes in a confined space and create clear class boundaries except for a few. Meanwhile, GCN and CCA-SSG methods can not capture higher-order information and show similar artifacts. TopoSRL manages to cluster some classes, but nodes in clusters are far from the center as in MPSN, which preserves more information. For example, the two clusters on the bottom and one from the right (corresponding to classes 1,2,3) are students from the same year but in different divisions. This information is preserved with TopoSRL as we can see three different clusters with some separation. However, there are a few overlaps as well, aligning with the nature of real-world data as students from the same year are often good friends (either due to extracurricular activities or study groups).

We present the comparison of TSNE plots for the senate-bills and contact-primary-school datasets in Figures S3 and S4. As we can see, the representations learned from supervised encoders such as MPSN, GCN, or self-supervised graph encoders trained with the CCA-SSG method show some artifacts while trying to classify the samples, as opposed to TopoSRL, which learns more sparse and distributed representations, conserving the topology.

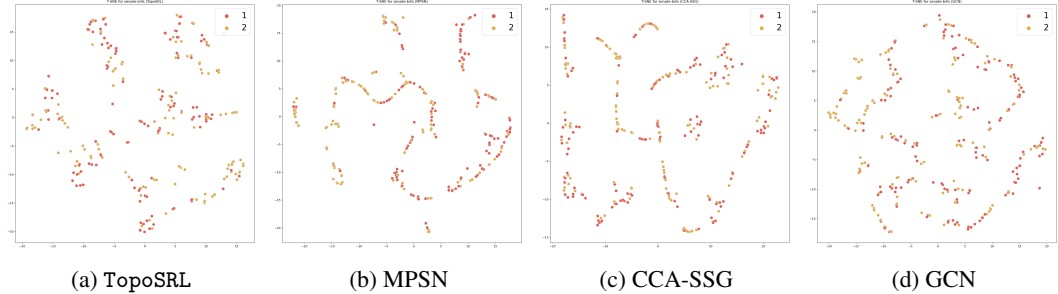

(a) TopoSRL     (b) MPSN     (c) CCA-SSG     (d) GCN

Figure S3: Comparison of TSNE plots of representations learned by various encoders for senate-bills dataset.

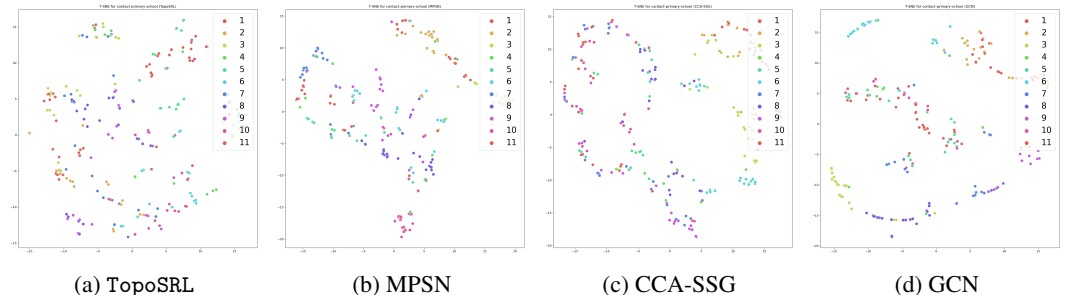

| (a) TopoSRL | (b) MPSN | (c) CCA-SSG | (d) GCN |

Figure S4: Comparison of TSNE plots of representations learned by various encoders `contact-primary-school` dataset.