# OpenReview forum: "TopoSRL: Topology preserving self-supervised Simplicial Representation Learning"
_NeurIPS.cc/2023/Conference — NeurIPS 2023 poster_

### Official Review · Reviewer_Lthj · 2023-07-02

**Soundness:** 2 fair
**Presentation:** 1 poor
**Contribution:** 3 good
**Rating:** 4
**Confidence:** 4

**Summary:**

The paper introduces a new self-supervised learning (SSL) method called TopoSRL for simplicial complexes. It claims to overcome the limitations of existing graph-based SSL methods by capturing higher-order interactions and preserving topology in learned representations. The authors propose a simplicial augmentation technique to generate enriched representations. They also propose a simplicial contrastive loss function to preserve local and global information in simplicial complexes. The experimental results allegedly demonstrate that TopoSRL outperforms state-of-the-art graph SSL techniques and supervised simplicial neural models on various datasets.

**Strengths:**

- The proposed method offers an interesting and intuitive approach.
- It addresses the capture of higher-order interactions and preserves topology in learned representations.
- The simplicial augmentation technique enriches representations.
- The proposed simplicial contrastive loss function preserves local and global information in simplicial complexes.

**Weaknesses:**

- The paper could improve its presentation and writing to enhance accessibility for general readers.
- The organization of the method section can be refined to provide clearer explanations and reduce overflow of mathematical expressions.
- Important definitions are placed in footnotes, which is not ideal.
- The effectiveness of the proposed method in practice is not adequately demonstrated.
- The choice of datasets used may not fully reflect the performance potential of the proposed method.

To enhance accessibility for general readers, improvements can be made in the presentation and writing style of the paper. Specifically, organizing the method section more effectively, providing clearer explanations, and reducing the overflow of mathematical expressions would greatly improve understanding. Placing important definitions within the main text instead of footnotes would also aid comprehension.

While the proposed method holds promise, the paper would benefit from further demonstration of its effectiveness in practical applications. The choice of datasets used in the experiments may not fully reflect the performance potential of the proposed method. Exploring a wider range of datasets, including those with a higher number of 3-simplices and higher-dimensional simplices, would help evaluate the method more comprehensively.

In terms of demonstrating the method's effectiveness, the results of the experiments are unclear. On one hand, the proposed method appears to improve the node-classification task, but it seems that 0-simplicies can already be readily incorporated into graphs. On the other hand, the authors mention that the method does not display outstanding performance in the experiment on simplicial closure datasets because it is not designed for a specific downstream task. This raises the question of what would be the proper downstream task that can truly showcase the benefits of the proposed method.

Therefore, it would be valuable to investigate the proper downstream task that can best showcase the advantages of the proposed method. Additionally, understanding how the method addresses the computational burden associated with higher-order simplices would provide valuable insights into its scalability.

**Questions:**

- It would be valuable to explore the proper downstream task that can showcase the true benefits of the proposed method.
- Additional information on the number of 3-simplices and higher-dimensional simplices in the datasets would help assess the effectiveness of the proposed method.
- It would be beneficial to understand how the method addresses the computational burden arising from higher-order simplices.

**Limitations:**

- The datasets used in the paper lack higher-order simplices, leading to overparameterization.
- The effectiveness of the proposed method may not be fully showcased without datasets containing a sufficient number of higher-order simplices.
- Addressing computational issues associated with higher-order simplices is important.
- If datasets with an adequate number of higher-order simplices are not readily available, it may be challenging for the proposed method to demonstrate its effectiveness in real-world data without access to extremely large datasets.

---

> ### Author Rebuttal · Authors · 2023-08-09
>
> We thank the reviewer for the valuable feedback. We address your concerns and questions as follows:
>
> ### Improving readability
> To improve the readability, we will place the important definitions in the main text, reduce the overflow of equations, and add an image to explain intuitively the working of the loss function.
>
> ### Choice of datasets and inclusion of higher order simplices
> We have chosen the standard simplicial complex datasets available used in the literature. It is common practice in the simplicial complex literature to not consider simplices of order four or above [1-5]. In fact, our results in Table 2 in the main manuscript align with this. Including higher-order information increases computational complexity without significant or no gains in performance at all. As such, it is a hyperparameter to restrict the simplicial complex to a certain order. Due to these reasons, we evaluate our method on datasets with 3-simplices and do not consider orders higher than that.
>
> ### Clarity on experiments and results
> $\texttt{TopoSRL}$ aims to learn task-agnostic representations without the requirements of any labels.
>
> We would like to clarify the confusion created by lines 284-290 in the manuscript. While $\texttt{TopoSRL}$ does not perform the best in the simplicial closure task, it has competitive results to a supervised MPSN. The reasoning is that the datasets for simplicial closure are skewed; therefore, the learned transformation functions $\phi$ are not expressive enough to predict simplicial closure. On the other, since $\texttt{TopoSRL}$ learns task-agnostic representations, it makes $\texttt{TopoSRL}$ more expressive than a supervised MPSN. This claim can be further verified by analyzing the confidence intervals of the best-performing methods in simplicial closure tasks.
>
> For example, on the email-Enron dataset, while SCNN performs the best in expectation, the confidence interval ranges from 0.53 to 0.69. While for $\texttt{TopoSRL}$, it ranges from 0.54 to 0.64. Similarly, the best-performing method on contact-high-school has a confidence interval of 0.27 to 0.67. While for $\texttt{TopoSRL}$, it is 0.43 to 0.43. This shows that even though $\texttt{TopoSRL}$ has sub-optimal performance for simplicial closure tasks, it learns expressive representations and is more reliable than its supervised counterparts.
>
> ### Effectiveness of $\texttt{TopoSRL}$ and Proper downstream tasks
> Since $\texttt{TopoSRL}$ is intended to be an SSL representation learning method for simplicial complexes, the proper downstream tasks are various tasks that exist in the simplicial complex literature. We have shown the capabilities of $\texttt{TopoSRL}$ on two such tasks, node classification and simplicial closure.
>
> Furthermore, we have conducted experiments for two more downstream applications, namely, 1) graph classification and 2) trajectory prediction. We have also presented the performance of partially labeled data when inferred with $\texttt{TopoSRL}$ and other baselines. These results are available in the global response and global response PDF. Please refer to **[GC], [TP],** and **[PLD]** in the global response where we highlight the effectiveness of $\texttt{TopoSRL}$.
>
> ### Computational Cost
> As mentioned in the manuscript, the cost of finding open simplices of order k is $\mathcal{O}(N_k^2)$. This is required to be done once outside of the training loop. Using the proposed augmentation technique, the complexity of adding random open simplices is $\mathcal{O}(N^o_k)$ where $N^o_k$ is the number of open simplices of order k. The complexity of one forward pass for one layer of an MPSN is $\mathcal{O}(D(N_{k-1} + N_{k} + N_{k+1}))$ for an arbitrary order k where D is the diameter of the underlying graph. Hence, calculating one forward pass costs $\mathcal{O}(2D(\sum_k N_k))$. Complexity to calculate $\mathcal{L}$ is $\mathcal{O}(DN_k^2+\delta^2)$ for any arbitrary order k, and $\delta$ is the number of sampled pairs for $\mathcal{L}_{rel}$ calculation. So the total complexity of the system is of $\mathcal{O}(LD(\sum_k N_k^2) + \sum_k N^{o}_k + K\delta^2)$, where $L$ is the number of layers in the encoder. So the complexity of $\texttt{TopoSRL}$ scales polynomially with the number of simplices in the $k^{th}$ order and the order of the simplicial complex $K$.
>
> [1] Christopher Wei Jin Goh, Cristian Bodnar, and Pietro Liò. Simplicial Attention Networks. 2022.DOI: 10.48550/ARXIV.2204.09455. URL: https://arxiv.org/abs/2204.09455.
>
> [2] Stefania Ebli, Michaël Defferrard, and Gard Spreemann. “Simplicial neural networks”. In:arXiv preprint arXiv:2010.03633 (2020).
>
> [3] Eric Bunch et al. “Simplicial 2-complex convolutional neural nets”. In: arXiv preprint arXiv:2012.06010 (2020).
>
> [4] Cristian Bodnar et al. “Weisfeiler and lehman go topological: Message passing simplicial
> networks”. In: International Conference on Machine Learning. PMLR. 2021, pp. 1026–1037.
>
> [5] Roddenberry, T.M., Glaze, N. and Segarra, S., 2021, July. Principled simplicial neural networks for trajectory prediction. In International Conference on Machine Learning (pp. 9020-9029). PMLR.

---

> > ### Comment · Reviewer_Lthj · 2023-08-20
> >
> > Thank you for the rebuttal. In my opinion, the reason 1-cycle features are the most discriminative is simply because of their prevalence. If a dataset contains more higher-order features relative to 1-cycles, including higher-order information should enhance performance. However, I agree that computing persistent homology for higher-order features on large datasets is computationally intractable, which evidently limits the practical consideration of 3-simplices as the most feasible option.

---

> > > ### Author Response · Authors · 2023-08-21
> > >
> > > Thank you for your response and feedback.
> > >
> > > Indeed, if there are sufficient number of higher-order open simplices in the dataset, including such higher-order simplices will increase the discriminative power. Although we validate our model on various tasks and simplicial complexes with up to 3-simplex (i.e., tetrahedrons), which highest-order of the simplex to choose is dataset dependent, but not a limitation of the TopoSRL model.
> > >
> > > To further clarify, regarding computation, finding an open/closed simplex of order $k$ is a one-time exercise and is not performed during training or inference. This is performed using standard tools.
> > >
> > > Finally, to the best of our knowledge, there are no other SSL methods for datasets beyond 0-simplices. As can be seen from the results in the global response and the manuscript, TopoSRL is beneficial for 0-simplex (e.g., node-level) SSL tasks in datasets with/without higher-order features, in addition to other higher-order simplex tasks such as simplicial closure or trajectory prediction.
> > >
> > > We hope this clarifies and addresses your concerns.

---

### Official Review · Reviewer_3cmn · 2023-07-06

**Soundness:** 3 good
**Presentation:** 3 good
**Contribution:** 3 good
**Rating:** 5
**Confidence:** 3

**Summary:**

This paper proposed a novel self-supervised learning (SSL) method for simplicial complexes. Advanced results are achieved by introducing appropriate augumentation methods and contrastive loss, taking into account the topological properties of the simplicial complex. It also compares the proposed method with the supervised and SSL methods, including the conventional method for graphs, and shows the effectiveness of the proposed method.


**Strengths:**

This paper proposes a novel SSL framework for simplicial complexes. The key to SSL, data augmentation and contrastive loss, are analogous to general methods, but they are non-trivial because they require consideration of data-specific characteristics, so it is commendable that a specific framework has been developed.

**Weaknesses:**

The concern is what effect the proposed method is intended to have. The experiments section includes a comparison of graph analysis methods and simplical complexes analysis methods and a comparison of supervised and SSL. Some of the references [1-6] include those where the effect is formally certified, but the target seems to be those that are treated as simplical complexes(e.g. polygons with holes). In other words, it is an object that cannot be processed as a graph.
Therefore, the comparison of the graph method with the simplical complexes method reiterates the superiority of the simplical complexes method, but since the data sets used are not common in graph analysis and are few in number, there is a strong suspicion that the superior data sets have been selected. On the other hand, if the subject is to propose SSL methods for simplical complexes, the main focus is on comparisons within simplical complexes. However, since supervised and SSL cannot be simply compared, it should be clarified how the effect should be recognized.

**Questions:**

Please give me your opinion on what you stated in Weakness.

**Limitations:**

The authors do not explicitly address Limitation. The conditions of applicability should be clarified.

---

> ### Author Rebuttal · Authors · 2023-08-09
>
> We appreciate the reviewer's insightful comments.
>
> ### On comparison with graph analysis methods
> While it is true that simplicial complexes cannot be directly processed as graphs, it is important to note that every simplicial complex inherently includes an underlying graph that captures the 0-simplices and 1-simplices. The comparison with graph analysis methods was motivated by the node classification task, an area where graph neural networks have shown significant advancements. By comparing results in node classification between $\texttt{TopoSRL}$ and graph analysis methods, we aim to highlight the potential value of $\texttt{TopoSRL}$, even in tasks traditionally handled by graph neural networks. The comparison with graph analysis methods further reinforces that including higher-order information improves performance, even in tasks at which graph neural networks are proficient.
>
> ### Regarding Dataset selection
> Our choices were not biased toward datasets with superior performance. Instead, we use the standard simplicial complex datasets representing real-world applications and enabling meaningful comparisons. Furthermore, to address reviewer RqW1's comment, we extended our experiments to include graph classification tasks on the TUDatasets. The results demonstrate that $\texttt{TopoSRL}$ can perform on par with supervised graph baselines and simplicial baselines. Other than this, $\texttt{TopoSRL}$ also outperforms or performs on par with graph SSL baselines, showing benefits on a standard graph dataset compared to graph SSL methods. For more details, please refer to **[GC]** in the global response.
>
> ### Comparison between supervised and SSL method
> The reason for comparison with supervised methods is to provide insights into how well the SSL method works compared to the supervised method. This comparison builds confidence in the expressive capabilities of $\texttt{TopoSRL}$ and solidifies its usefulness in learning simplex representations in settings with less-labeled or unlabeled data.
>
> We hope you appreciate our responses and that all your concerns are addressed.

---

> > ### Comment · Reviewer_3cmn · 2023-08-12
> > **Thank you for your clarification**
> >
> > Let me confirm one point: I understand that simplical complex can also be formally analyzed as a graph. However, previous studies have shown that analysis as a graph is often inadequate because it loses information on the simplical complex. The purpose of this paper is an extension to SSL, and comparisons with graph analysis methods no longer seem essential. Is the comparison with the graphical analysis method posted to clarify that the graphical analysis method is not sufficient to analyze the simplical complex in the SSL setting as well? Or do you consider them as a baseline? I would like to see these points clarified. Otherwise, the suspicion remains that they are trying to make their effectiveness look better (outperforming numerous methods) with unfair comparisons.

---

> > > ### Author Response · Authors · 2023-08-13
> > > **Thank you for the question**
> > >
> > > Yes, the purpose is to clarify and show that in the SSL setting as well, the proposed simplicial SSL method (i.e., TopoSRL) performs better than graph SSL methods, as demonstrated in the existing supervised simplicial representation learning methods, e.g., MPSN [1]. This result might not be immediately evident in the SSL setting as it depends on the augmentation technique and contrastive loss, which are central to our proposal in TopoSRL. Also, we re-emphasize that there are no simplicial SSL methods (to the best of our knowledge) that one can readily use as a baseline.
> > >
> > > We hope that our responses clarify the reviewer's concerns.
> > >
> > > [1] Cristian Bodnar et al. “Weisfeiler and Lehman go topological: Message passing simplicial networks”. In: International Conference on Machine Learning. PMLR. 2021, pp. 1026–1037.

---

> > > > ### Comment · Reviewer_3cmn · 2023-08-14
> > > > **Re:**
> > > >
> > > > I expect the presentation to be revised. Since this is a subject that has not yet been discussed much, I think it is inevitable that there are few comparisons to be made. I plan to review the original paper and comments again and re-evaluate.

---

> > > > > ### Author Response · Authors · 2023-08-16
> > > > > **Thank you**
> > > > >
> > > > > Thank you for your time and effort. We will further clarify these aspects in the revised paper.

---

> > > > > > ### Comment · Reviewer_3cmn · 2023-08-19
> > > > > >
> > > > > > My concerns were appropriately addressed and I have a positive impression regarding the details. I am positive on the content, but will keep the score as it is difficult to judge the impact on the conference due to the niche area. I defer to the AC on the decision, including the impact on the conference.

---

### Official Review · Reviewer_VjqH · 2023-07-06

**Soundness:** 2 fair
**Presentation:** 3 good
**Contribution:** 2 fair
**Rating:** 6
**Confidence:** 3

**Summary:**

The paper introduces a new algorithm for self-supervised learning for simplicial complex-based networks.
The paper proposes a new simplicial contrastive loss function that contrasts the generated simplices to preserve local and global information present in the simplicial complexes.
Experiments show the utility of the proposed method. In particular, it outperforms another SSL method for simplicial complexes. Also, its quality is on par with supervised counterparts.

**Strengths:**

The paper belongs to a new field: topological deep learning, which is a generalization of graph neural networks to more complex structures (simplicial complexes, cell complexes, hypergraphs, etc). There are not so many papers dedicated to this field. Literature review is adequate.
Experiments show that the proposed SSL method outperforms another (SOTA) SSL method. Confidence intervals are provided. Theoretical results regarding the proposed losses are provided. The quality is almost on par with supervised methods.
he language of the manuscript is clear.

**Weaknesses:**

Learning with simplical complexes is a quite niche research area. So, the impact of the paper is limited.
Also, main ideas like using augmented complexes is similar to such augmentation for graphs [1] and the contrastive loss is also quite popular.
Possible applications of simplicial networks must be discussed in more details.

[1] Kaveh Hassani and Amir Hosein Khasahmadi. Contrastive multi-view representation learning
429 on graphs. In ICML, pages 4116–4126, 2020.

**Questions:**

1. Can you please explain the Fig.2, right. What is the structure of the dataset? What is the prediction target?
2. The inter-view cost matrix is used to minimize the distance between the representation of a simplex and an aggregate representation of a sub-simplicial complex surrounding this simplex in the other augmented simplicial complex.
Why this value should be minimized?
3. Why you haven't compared with other SSL baselines in Table 3?

**Limitations:**

Authors adequately addressed the limitations.

---

> ### Author Rebuttal · Authors · 2023-08-09
>
> We thank the reviewer for the valuable feedback. We address your concerns and questions as follows:
>
> ### On similarity with graph augmentation and contrastive learning
> We acknowledge that the concept of augmentation for graphs has gained popularity in graph SSL methods. However, as highlighted in the Related Works section and supported by empirical evidence in Table 2 in the main manuscript, merely extending graph augmentation methods yields unsatisfactory results on considered tasks. Furthermore, as presented in Table S2 in supplementary material, both the terms involved in the proposed contrastive loss are essential. Although we drew inspiration from the general graph SSL pipeline, adapting it to simplicial complex data posed non-trivial challenges such as requiring complex augmentation techniques, computational cost from negative sampling algorithms based, or requiring components to avoid degenerative solutions empirically. A detailed discussion of these challenges is available in the Introduction section from lines 34-53 in the manuscript.
>
> ### Possible applications
> We appreciate your feedback. There are quite a few applications [4] of simplicial complexes, image processing [1], molecular structure analysis [2], trajectory prediction [3], etc.
>
> In response to Reviewer Lthj, we assessed $\texttt{TopoSRL}$ on the trajectory prediction task and presented the findings in Table G3a within the global response PDF. The results indicate that $\texttt{TopoSRL}$ outperforms SCNN and ScoNe [3] when applied to the Ocean Drifters dataset. When tested on synthetic data, $\texttt{TopoSRL}$ demonstrates performance on par with a supervised SCNN. These results further highlight the expressive representation capabilities of $\texttt{TopoSRL}$ on oriented simplicial complexes and its use cases in practical applications.
>
> Please refer to **[TP]** in the global response for more details.
>
> ### More explanation on Figure 2
> Figure 2 compares T-SNE plots for four different methods: $\texttt{TopoSRL}$, CCA-SSG, MPSN and GCN. The dataset used is contact-high-school simplicial complex, and the node classes are the prediction target. Each point in the plot corresponds to the representations of a node from the contact-high-school dataset. To restate what is mentioned in the manuscript, MPSN can cluster most nodes in a confined space and create clear class boundaries except for a few. Meanwhile, GCN and CCA-SSG methods cannot capture higher-order information and show similar artifacts. $\texttt{TopoSRL}$ manages to cluster some classes, but nodes in clusters are far from the center as in MPSN, which preserves more information. For example, the two clusters on the bottom and one from the right (corresponding to classes 1,2,3) are students from the same year but in different divisions. This information is preserved with $\texttt{TopoSRL}$ as we can see three different clusters with some separation. However, there are a few overlaps as well, aligning with the nature of real-world data as students from the same year are often good friends (either due to extracurricular activities or study groups).
>
> ### Importance of minimizing distance between simplex and sub-simplial complex
> Proposition 2 demonstrates that minimizing the proposed loss function simultaneously maximizes the mutual information between simplices and their corresponding representations. On the other hand, Proposition 1 reveals that minimizing the proposed loss function leads to minimizing the distance between a simplex's representation and the aggregate representation of a sub-simplicial complex surrounding that simplex in the other augmented simplicial complex. This process also maximizes mutual information (MI), aligning with the principles seen in earlier methods like DGI, GCA, and CCA-SSG. However, unlike DGI and GCA methods, $\texttt{TopoSRL}$ achieves MI maximization without the need for additional components such as inductive readout functions and discriminator functions or the need for negative samples. As a result, $\texttt{TopoSRL}$ maximizes MI drastically while being computationally efficient.
>
> ### Lack of graph baselines in simplicial closure task
> Simplicial closure is a task inherently specific to simplicial complexes and is not well-suited for a graph encoder. Due to this, the results presented in the manuscript do not include a direct comparison between simplicial closure and graph-based baselines. Instead, our focus has been on evaluating the performance of simplicial closure using appropriate methods tailored for simplicial complex tasks.
>
> ### Impact
> Thank you for your response. As such, representation learning for graphs and higher-order generalizations such as hypergraphs, simplicial, and cell complexes is a timely topic receiving much attention in the community. The method is developed for simplicial complexes but has implications beyond simplicial complexes. It can inspire more developments in representation learning for other complex data structures, such as hypergraphs or multi-relational data. Since annotating simplices of different orders or labeling a simplicial complex is difficult, developing and understanding SSL methods for simplicial complexes is natural.
>
> [1] Asao, Y., Nagase, J., Sakamoto, R. and Takagi, S., 2021. Image recognition via Vietoris-Rips complex. arXiv preprint arXiv:2109.02231.
>
> [2] Gong, W., Wee, J., Wu, M.C., Sun, X., Li, C. and Xia, K., 2022. Persistent spectral simplicial complex-based machine learning for chromosomal structural analysis in cellular differentiation. Briefings in Bioinformatics, 23(4), p.bbac168.
>
> [3] Roddenberry, T.M., Glaze, N. and Segarra, S., 2021, July. Principled simplicial neural networks for trajectory prediction. In International Conference on Machine Learning (pp. 9020-9029). PMLR.
>
> [4] Hajij, Mustafa, Ghada Zamzmi, Theodore Papamarkou, Nina Miolane, Aldo Guzmán-Sáenz, and Karthikeyan Natesan Ramamurthy. "Higher-order attention networks." arXiv preprint arXiv:2206.00606 (2022).

---

> > ### Comment · Reviewer_VjqH · 2023-08-20
> > **Response**
> >
> > Thank you for the detailed clarification. You should improve from manuscript accordingly. Since my questions are addressed, I'm raising my score.

---

> > > ### Author Response · Authors · 2023-08-21
> > > **Thank you**
> > >
> > > Thank you for your time and effort. We will improve the manuscript and include all the clarifications.

---

### Official Review · Reviewer_T7H9 · 2023-07-07

**Soundness:** 2 fair
**Presentation:** 1 poor
**Contribution:** 2 fair
**Rating:** 4
**Confidence:** 3

**Summary:**

This work designs an SSL pipeline coined as TopoSRL for simplicial complex data, and proposes a new contrastive loss function to preserve topology information to learn more expressive representations.

**Strengths:**

1. The motivation is meaningful.
2. The experiments are fairly comprehensive.

**Weaknesses:**

1. In Table 1 and  Table  3, the baselines are all supervised settings, i.e., lack of comparison with other self-supervised algorithms.
2. The visualization results in Figure 2 do not adequately reflect the differences in clustering ability among the algorithms. Can authors give some numerical evaluations such as AMI and ARI?

**Questions:**

In lines 131-132,  the statements about lower-adjacent neighbors and upper-adjacent neighbors are not clear enough.  Can authors give some examples like Figure 1?
In lines 137-138, What does the \phi mean? And the statements like "∀τk+1 ∈ C(σk)" need to move outside.

**Limitations:**

N/A.

---

> ### Author Rebuttal · Authors · 2023-08-09
>
> We thank the reviewer for the valuable feedback. We address your concerns and questions as follows:
> ### Lack of SSL baselines
> To compare with more graph SSL baselines, we have added BGRL and GCA as graph SSL baselines and GIN [1] as graph supervised baseline. The results are available in Table G1 in the global response PDF. The graph baselines do not perform as well as simplicial baselines. Whereas, $\texttt{TopoSRL}$ outperforms all the graph baselines on the node classification task and is competitive with supervised simplicial neural network baselines.
>
> Please refer to **[GBS]** in global response for more details.
> ### AMI and ARI
> We report the AMI and ARI metrics for contact-high-school dataset presented in Figure 2 from the manuscript in the following table:
> |      |  AMI   |  ARI   |
> |------|--------|--------|
> | $\texttt{TopoSRL}$ | <u>0.7608</u> | <u>0.7094</u> |
> | CCA-SSG | 0.6003 | 0.3965 |
> | MPSN   | **0.7811** | **0.7498** |
> | GCN    | 0.1634 | 0.1049 |
>
> As we can see by these metrics and as mentioned in the manuscript, MPSN achieved better clustering compared to $\texttt{TopoSRL}$. However, MPSN also loses some expressiveness over $\texttt{TopoSRL}$ as all the nodes in a cluster are in a confined space and lose information. In contrast, $\texttt{TopoSRL}$ manages to cluster most classes but the nodes in a cluster are not too close like in MPSN. For more details, please refer to lines 302-310 in the manuscript.
> ### Upper and lower adjacent neighbors
> Thank you for the question. In Figure 1b in the manuscript, in the original view, the 2-simplices (or filled triangles) indicated by {Sanitizer, Wipes, Shampoo} and {Sanitizer, Wipes, Detergent} are lower adjacent neighbors as they share a common lower order 1-simplex (or edge) {Sanitizer, Wipes}. At the same time, 1-simplices {Sanitizer, Wipes} and {Sanitizer, Detergent} are upper adjacent as they are both part of a higher order 2-simplex {Sanitizer, Wipes, Shampoo}. Additionally, 2-simplices {Sanitizer, Wipes, Shampoo} and {Sanitizer, Wipes, Detergent} are also upper adjacent neighbors as they are both part of 3-simplex {Sanitizer, Wipes, Shampoo, Detergent}.
> ### What is $\phi$?
> $\phi$ is a message transformation function. In our setting, we use a one-layer MLP with no bias.
> ### Correcting notation
> Thank you for pointing this out. The RHS of eqautions in line 138 should be $AGGREGATE(\phi_C(h^t_{\sigma_k},h^t_{\tau_{k+1}}),\forall \tau_{k+1}\in C(\sigma_k))$. This will be corrected in the revised paper.
>
> We hope you appreciate our responses and that all your concerns are addressed.
>
> [1] Xu, Keyulu, Weihua Hu, Jure Leskovec, and Stefanie Jegelka. "How powerful are graph neural networks?." arXiv preprint arXiv:1810.00826 (2018).

---

> > ### Comment · Reviewer_T7H9 · 2023-08-16
> >
> > Thank you for providing the rebuttals.
> >
> > I noticed that this work did not mention graphMAE [1], a typical graph self-supervised learning method,  in the manuscript and the global response PDF. I would like to see the advantages of the method proposed in this work over GraphMAE.
> >
> > [1] Zhenyu Hou, Xiao Liu, Yukuo Cen, Yuxiao Dong, Hongxia Yang, Chunjie Wang, Jie Tang: GraphMAE: Self-Supervised Masked Graph Autoencoders. KDD 2022: 594-604

---

> > > ### Author Response · Authors · 2023-08-17
> > > **Thank you for pointing us to this reference.**
> > >
> > > Thank you for pointing us to this reference. We will refer to it in the revised paper.
> > >
> > > GraphMAE is indeed an alternative approach that uses a reconstruction loss rather than the commonly used contrasting loss in the graph-based SSL methods. However, GraphMAE is not designed for simplicial complexes or topological datasets, and extending it to simplicial complexes is a study in its own right, e.g., if we mask a simplex of order $k$, should we mask all the sub-simplices created by this complex or how to regularize the reconstruction loss to account for global information in the simplicial complex. As we have seen in Table S5 in the supplementary material, using both local and global information yields more expressive representations.
> > >
> > > Nonetheless, we present below a comparison of GraphMAE with TopoSRL in the following table. While GraphMAE has a better performance compared to CCA-SSG (as noted in the suggested reference), the conclusions in the manuscript about TopoSRL remain the same, i.e., GraphMAE (like other graph-based models) do not perform well compared to simplicial-based models and TopoSRL.
> > >
> > > | Method    | contact-high-school | contact-primary-school | Senate-bills |
> > > |-----------|-------------|----------------|--------------|
> > > | CCA-SSG   | 0.68 ± 0.16   | 0.14 ± 0.07      | 0.62 ± 0.04    |
> > > | GraphMAE  | 0.78 ± 0.05 | 0.2 ± 0.02     | 0.57 ± 0.01  |
> > > | GIN       | 0.18 ± 0.04   | 0.16 ± 0.02      | 0.53 ± 0.04    |
> > > | MPSN      | 0.89 ± 0.01 | 0.79 ± 0.06    | 0.75 ± 0.05  |
> > > | TopoSRL   | 0.92 ± 0.05 | 0.61 ± 0.05    | 0.72 ± 0.06   |
> > >
> > > We hope that this response clarifies your concern.

---

### Official Review · Reviewer_RqW1 · 2023-07-21

**Soundness:** 3 good
**Presentation:** 4 excellent
**Contribution:** 2 fair
**Rating:** 6
**Confidence:** 4

**Summary:**

The work proposes a Self-Supervised Learning (SSL) approach for simplicial complexes, extending graph SSL to higher-order structures. While recent works exist that extend and adapt standard graph representation learning techniques to simplicial complexes, none consider SSL. In particular, the authors propose to employ a contrastive learning procedure that involves two steps: first, for each complex, two artificial simplicial complexes are created with a data augmentation technique that randomly removes closed simplices and randomly adds open ones to the complex. Then, the complexes are contrasted with an ad hoc contrastive loss that has two goals: the first is to minimize the distance of each k-simplex across the two augmented views, and the second is to minimize the distance of a $k$-simplex and the aggregated representation of a sub-simplicial complex surrounding it in the other view. The per-simplex distances are simply given by the squared euclidean distance. The authors then prove that the proposed contrastive loss implicitly maximizes the mutual information between a simplex and its neighborhood within the same augmented simplex and the mutual information between a simplex and its neighbors in the other augmented complex. The approach is tested over node classification and simplicial closure for a set of social network datasets, and shows competitive performance wrt a set of supervised baselines both acting on graphs and simplicial complexes. The work is also shown to outperform an existing graph SSL baseline.

**Strengths:**

- The paper is well written and easy to understand, providing the reader the necessary background and explaining the contribution in simple terms.
- The work is the first to study the problem of self-supervised learning on simplicial complexes, noting that existing graph SSL methods may not be directly adapted. Given the recent raise of interest in the field, this work may well be the first of many.
- The approach makes intuitive sense, is simple and has theoretical justification. Being model agnostic, the approach can (and has been) used on any simplicial complex encoder.
- The experimental evidence seems to indicate that, on the considered tasks, the approach is surprisingly competitive with supervised baselines.

**Weaknesses:**

- Part of the reason why there are no works on self-supervised learning on simplicial complexes is that data distributions that can be naturally designed as simplicial complexes are rare, and the work indeed only considers social network datasets that are only seen in works of the same niche. For this reason, it would be interesting to see whether the approach still shows benefits on a standard graph dataset converted to complex with lifting technique (e.g. using clique complexes). Indeed, main works in the field, e.g. [1], usually also compare on established graph datasets such as TUDatasets.
- The method only presents a single graph SSL baseline, assessing it to be the state of the art. In general, it would still be desirable to have several baselines for two reasons: i) the considered SOTA model could prove to be better only in the datasets, tasks or evaluation setting considered in the original paper, and ii) the comparison with less-performing baselines would still be valuable to the experimenter as well as the reader.
- It is not immediately clear how the supervised graph-learning baselines were chosen, and why no chosen network is at least 1-WL expressive. It would be good to at least report the results for GIN [2].
- The method and insights may not be of immediate interest to the broader NeurIPS community as they are specifically tailored to simplicial complexes, and would probably be of greater interest in a graph-specific venue. While the venue also accepts domain-specific works, these usually have more general insight that can be appreciated by practitioners outside the niche.
- Much of the utility of self-supervised learning comes when it can be used for pretraining an encoder on a large dataset that can be later fine-tuned for a downstream task. It would be great to see if this is the case for the proposed approach, as otherwise it's not immediately clear why it would be advisable wrt to standard supervised models.

Overall, I am more than happy to raise my score if my concerns are addressed.

**References**:

[1] Bodnar, C., Frasca, F., Wang, Y., Otter, N., Montufar, G. F., Lio, P., & Bronstein, M. (2021, July). Weisfeiler and lehman go topological: Message passing simplicial networks. In International Conference on Machine Learning (pp. 1026-1037). PMLR.

[2] Xu, K., Hu, W., Leskovec, J., & Jegelka, S. (2018). How powerful are graph neural networks?. arXiv preprint arXiv:1810.00826.

**Questions:**

- Table 2: to be called an ablation study, shouldn’t it also consider the case in which the closed simplices are not removed while the open simplices are still added?
- Are the results averaged over different seeds? how many runs were performed?

---

> ### Author Rebuttal · Authors · 2023-08-09
>
> We thank the reviewer for the valuable feedback. We address your concerns and questions as follows:
> ### Graph classification on TUDatasets using clique lifting
> We have incorporated your suggestion and included Table G2 in the global response PDF, presenting the performance comparison between the proposed method $\texttt{TopoSRL}$ with the graph and simplicial baselines on graph classification datasets.
>
> As we can see in Table G2, $\texttt{TopoSRL}$ performs on par with supervised graph baselines and simplicial baselines. Further, $\texttt{TopoSRL}$ outperforms or performs on par with graph SSL baselines, showing the advantages of $\texttt{TopoSRL}$ on a standard graph dataset with clique lifting compared to graph SSL methods.
>
> For more details, please refer to **[GC]** in the global response.
>
> ### Graph baselines
> To compare with more graph SSL baselines, we have added BGRL and GCA as graph SSL baselines with GIN as graph supervised baseline. The results are available in Table G1 in the global response PDF.
>
> The graph baselines do not perform as well as simplicial baselines. Whereas, $\texttt{TopoSRL}$ outperforms all the graph baselines on the node classification task and is competitive with supervised simplicial neural network baselines.
>
> Please refer to **[GBS]** in the global response for more details.
>
> ### Transfer learning/Fine-tuning
> In graph SSL, generally, the encoder is trained on a graph that it is supposed to infer on, unlike in images where a large network can be fine-tuned to a domain-specific dataset later. However, to show the expressiveness of the representation of our method, we perform and present an experiment with partially labeled data. The results for the same are available in Table G4 in the global response PDF.
>
> $\texttt{TopoSRL}$ significantly improved performance by about 5% over supervised MPSN in the 20-80 and 40-60 split. This provides empirical evidence about the expressive capabilities of $\texttt{TopoSRL}$ and its efficacy with less-labeled data. Hence, $\texttt{TopoSRL}$ would be preferable over standard supervised models in the less-labeled data setting.
>
> Please refer to **[PLD]** in the global response for more details about the experiment setup.
>
> ### Open-only augmentation
> Adding open simplices is a better augmentation technique than random augmentation, but its performance deteriorated from the proposed augmentation technique. This occurs because the contrastive objective function is more effective when more information to contrast is present. Removing closed simplices allows the contrastive loss and encoder to contrast more information, resulting in higher performance instead of only adding open simplices.
>
> Please find our answer in **[OOA]** in global response as a response to this question.
>
> ### Seeds and runs
> We follow the standard practice where all the results are averaged over ten different seeds, and one run is performed for each seed.
>
> ### Interests to the NeurIPS community and impact
> We understand that the NeurIPS community has a broader spectrum of interest and that there are many graph representation learning works that were reported to NeurIPS. We believe that our new results on SSL and the importance of topology preservation have broader relevance. The proposed method has implications beyond simplicial complexes and can inspire more developments in representation learning for other complex data structures, such as hypergraphs or multi-relational data. As such, developing representation learning networks for higher-order networks is a timely topic gaining significant attention.
>
> We hope you appreciate our responses and that all your concerns are addressed.

---

> > ### Comment · Reviewer_RqW1 · 2023-08-17
> >
> > I thank the authors for their rebuttal, which addresses most of my questions and concerns.
> >
> > I am happy to see that the approach also works on clique-lifted graphs, as this increases its general usefulness. I am also glad to see more comparisons with existing baselines and that the method benefits partially labelled scenarios. Moreover, it is good to see that the open-only augmentation yields benefits over the random augmentation but not as much as the full augmentation, as it was expected.
> > I thank the authors for clarifying the experimental setting, having expressed the number of seeds and runs.
> > However, I still believe the work to require some more general insight for the broader community to be fully interested.
> >
> > Carefully considering these aspects, I would like to raise my score to a weak accept.

---

> > > ### Author Response · Authors · 2023-08-18
> > > **Thank you**
> > >
> > > Thank you for the positive and constructive feedback. We will add additional insights on the results and the method (including feedback from other reviewers) to make the work accessible to a broader audience.

---

### Author Rebuttal · Authors · 2023-08-09

We thank all the reviewers for their insightful comments and thoughtful questions. As suggested by reviewers, we have conducted the following experiments and presented them in the global response PDF. The new results are in red color.

### **Graph Classification on TU Datasets [GC]**
- As suggested by reviewer RqW1, we have included new experiments to provide insights on how well $\texttt{TopoSRL}$ works on standard graph datasets with clique lifting. To this end, we have conducted experiments on graph classification task on Proteins, NCI1, Reddit-B, Reddit-M, and IMDB-B datasets from the TUDatasets repository [2]. To extract graph embeddings from simplex embeddings, we perform average pooling on the embeddings of each order $k$ as $z^i_k$, where $i$ denotes $i^{th}$ graph and the graph embeddings are the concatenation of all the orders as follows: $z^i = [z^i_0|| z^i_1|| …|| z^i_K]$ where $K$ is the maximal order of the simplex in the graph and $||$ is the concatenation operator. Due to the time limitation, we have only included results for one seed.
- As we can see in Table G2, $\texttt{TopoSRL}$ performs on par with supervised graph baselines and simplicial baselines. Further, $\texttt{TopoSRL}$ outperforms or performs on par with graph SSL baselines, showing the advantages of $\texttt{TopoSRL}$ on a standard graph dataset with clique lifting compared to graph SSL methods. Nonetheless, we can include results from more seeds in the revised paper.


### **More Graph SSL baselines [GBS]**
- As suggested by most of the reviewers, we have added three more graph baselines for comparison. In particular, our analysis has added GCA and BGRL as graph SSL baselines and GIN as supervised graph baselines. All the graph SSL baselines were trained with GIN as an encoder for both graph and node classification. Due to time constraints, we have only added the two graph SSL baselines. However, simplicial baselines still outperform the newly added graph baselines, and the results will also hold for other graph baselines.
- The results for node classification with new baselines are available in Table G1 in the global response. As we can see, the graph baselines do not perform as well as simplicial baselines. Whereas $\texttt{TopoSRL}$ outperforms all the graph baselines on the node classification task and is competitive with supervised simplicial neural network baselines.

### **Partially labeled data [PLD]**
- To test the ability of $\texttt{TopoSRL}$ to learn expressive representations, we perform the following experiment, 1) Use the pre-trained $\texttt{TopoSRL}$ encoders to extract representations and 2) Use only a partial labeled (e.g., 20% train and 80% test, 40% train and 60% test, etc.) data to train a logistic regression classifier for node classification task in the contact-high-school dataset. MPSN and GCN are trained with cross-entropy loss on the train set.
- Since the weights for the $\texttt{TopoSRL}$ encoder trained without labels have been saved, no new encoders were trained to produce the results. The results for the same are presented in Table G4 in the global responses PDF.
- As we increase the size of the train set, the performance increases across all the methods. Furthermore, $\texttt{TopoSRL}$ has a significantly improved performance of about 5% over supervised MPSN in the 20-80 and 40-60 split. This provides empirical evidence about the expressive capabilities of $\texttt{TopoSRL}$ and its efficacy with less-labeled data. Hence, $\texttt{TopoSRL}$ would be preferable over standard supervised models in the less-labeled data setting.

### **Open-only simplex argumentation [OOA]**
- As suggested by reviewer RqW1, we now considered a case of augmentation where only open simplices are added while close simplices are not removed. We have conducted experiments with this method and presented the results in Table G3b of the global response PDF.
- Only adding open simplices is a better augmentation technique than random augmentation, but its performance deteriorates compared to the proposed augmentation technique. This occurs because the contrastive objective function is more effective when there is more information to contrast. Removing closed simplices allows the contrastive loss and encoder to contrast more information, resulting in higher performance instead of only adding open simplices.

### **Trajectory Prediction (additional task) [TP]**
- As suggested by reviewer LthJ, we perform experiments on the trajectory prediction task to demonstrate the effectiveness of $\texttt{TopoSRL}$ in downstream applications.
- The task of trajectory prediction (TP) is described formally as follows: A trajectory over a simplicial complex $\mathcal{X}$ with node set $\mathcal{V}$ is a sequence of nodes $[i_0, i_1, \dots, I_{p-1}]$ such that $ I_{j-1}$ and $i_j$ are adjacent. Now, TP aims to predict what node $i_p$ will be. For more details on trajectory predictions, please refer to [1].
- The results for the task are presented in Table G3a in the global response, where we compare our method with supervised ScoNe [1] and SCNN on ocean drifters and synthetically generated data using the approach proposed in [1].
- The results indicate that $\texttt{TopoSRL}$ outperforms SCNN and ScoNe [1] when applied to the Ocean Drifters dataset. When tested on synthetic data, $\texttt{TopoSRL}$ demonstrates performance on par with a supervised SCNN. These results further highlight the expressive representation capabilities of $\texttt{TopoSRL}$ on oriented simplicial complexes and its use cases in practical applications.

[1] Roddenberry, T.M., Glaze, N. and Segarra, S., 2021, July. Principled simplicial neural networks for trajectory prediction. In International Conference on Machine Learning (pp. 9020-9029). PMLR.

[2] Morris, C., Kriege, N.M., Bause, F., Kersting, K., Mutzel, P. and Neumann, M., 2020. Tudataset: A collection of benchmark datasets for learning with graphs. arXiv preprint arXiv:2007.08663.

---

### Decision · Program_Chairs · 2023-09-21

**Decision:**

Accept (poster)

**Comment:**

The paper presents a self-supervised method for simplicial complexes, demonstrating competitive performance over prior art over a range of tasks. The paper received borderline-positive reviews initially, and then underwent further discussion in the post-rebuttal phase. Key strengths were consistently identified by the reviewers in the quality of the exposition, the novelty of the problem (which could potentially become seminal in the graph learning commmunity), and the quality of the experimental results. The main weaknesses were given in terms of limited comparisons, to which the authors responded with an extensive rebuttal, including several additional experiments and comparisons as requested by the reviewers. As a consequence, two of the initial ratings were raised to Weak Accept. We believe the paper carries a good potential impact, and meets the quality bar for presentation at NeurIPS. In the light of the main strengths as recognized by the reviewers and the additional input in the authors' responses, we recommend acceptance with the additional recommendation to revise the paper accordingly for the camera ready version.